# Statistical characteristics of extreme daily precipitation during 1501 BCE – 1849 CE in the Community Earth System Model

Woon Mi Kim[1,2], Richard Blender[3], Michael Sigl[1,2], Martina Messmer[1,2], and Christoph C. Raible[1,2]

[1]Climate and Environmental Physics, Physics Institute, University of Bern, Switzerland
[2]Oeschger Centre for Climate Change Research, University of Bern, Switzerland
[3]Meteorological Institute and Center for Earth System Research and Sustainability (CEN), University of Hamburg, Germany
**Correspondence:** Woon Mi Kim (woonmi.kim@climate.unibe.ch)

**Abstract.** In this study, we analyze extreme daily precipitation during the pre-industrial period from 1501 BCE to 1849 CE in simulations from the Community Earth System Model version 1.2.2. A peak-over-threshold (POT) extreme value analysis is employed to examine characteristics of extreme precipitation and to identify connections of extreme precipitation with the external forcing and with modes of internal variability. The POT analysis shows that extreme precipitation with similar statistical characteristics, i.e., the probability density distributions, tends to cluster spatially. There are differences in the distribution of extreme precipitation between the Pacific and Atlantic sectors and between the northern-high and southern-low latitudes.

Extreme precipitation during the pre-industrial period is largely influenced by modes of internal variability, such as El Niño-Southern Oscillation (ENSO), the Pacific North American, and Pacific South American patterns among others, and regional surface temperatures. In general, the modes of variability exhibit a statistically significant connection to extreme precipitation in the vicinity to their regions of action.The exception is ENSO, which shows more widespread influence on extreme precipitation across the Earth. In addition, the regions where extreme precipitation is more associated either by a mode of variability or by the regional surface temperature are distinguished. Regional surface temperatures are associated with extreme precipitation over lands at the extratropical latitudes and over the tropical oceans. In other regions, the influences of modes of variability are still dominant. Effects of the changes in the orbital parameters on extreme precipitation are rather weak compared to those of the modes of internal variability and of the regional surface temperatures. Still, some regions in central Africa, southern Asia, and the tropical Atlantic ocean show statistically significant connections between extreme precipitation and orbital forcing, implying that in these regions, extreme precipitation has increased linearly during the 3351-year pre-industrial period. Tropical volcanic eruptions affect extreme precipitation more clearly in the short term up to a few years, altering both the intensity and frequency of extreme precipitation. However, more apparent changes are found in the frequency than the intensity of extreme precipitation. After eruptions, the return periods of extreme precipitation increase over the extratropical regions and the tropical Pacific, while a decrease is found in other regions. The post-eruption changes in the frequency of extreme precipitation are associated with ENSO, which itself is influenced by tropical eruptions.

Overall, the results show that climate simulations are useful to complement the information on pre-industrial extreme precipitation, as they elucidate statistical characteristics and long-term connections of extreme events with natural variability.

## 25  1  Introduction

Extreme daily precipitation, which often causes devastating flood events, is a difficult phenomenon to study due to its rare occurrence and short-lived nature. At regional scale, extreme precipitation events are caused by meso- and synoptic-scale processes (Pfahl and Wernli, 2012; Pfahl, 2014) and at global scale, they are influenced by large-scale modes of variability, such as El Niño-Southern Oscillation (ENSO) or the North Atlantic Oscillation (NAO) among others (Kenyon and Hegerl,
2010; Sun et al., 2015). In recent years, the topic has attracted more and more attention as the behavior of extreme precipitation is expected to change differently to that of mean precipitation in a warmer world (Allen and Ingram, 2002; Trenberth et al., 2003; Pall et al., 2007). While mean precipitation is expected to follow largely the "wet gets wetter, and dry gets drier" rule (Trenberth et al., 2003; Chou and Neelin, 2004; Chou et al., 2009), extreme precipitation is projected to increase over the entire globe (Trenberth et al., 2003; Donat et al., 2016; Fischer and Knutti, 2016). The "wet gets wetter, and dry gets drier" pattern
denotes the intensification of the global hydrological cycle, which is controlled by a tropospheric energy budget (Boer, 1993; Allen and Ingram, 2002; Yang et al., 2003). Nevertheless, it is also noted that this pattern for the future mean precipitation is more heterogeneous over land areas in observations and climate models (Roderick et al., 2014; Byrne and O'Gorman, 2015) and breaks in the tropics in CMIP5 models (Chadwick et al., 2013).

Unlike changes in mean precipitation which are largely regulated by the available energy budget, changes in extreme pre-
cipitation are constrained by the available maximum low-level atmospheric moisture at a given temperature following the Clausius–Clapeyron (C-C) relationship (Allen and Ingram, 2002; Pall et al., 2007). The reasoning is as follows: the low-level atmosphere can hold more moisture with increasing temperatures, which in turn leads to an increase in extreme precipitation (Trenberth et al., 2003; Pall et al., 2007; Fischer and Knutti, 2016). The rate of increase of extreme precipitation given by C-C is 6-7% per 1 degree of warming. This relationship holds mostly true over higher latitudes where the air is usually closer to
saturation and relative humidity is roughly constant (Allen and Ingram, 2002). Mean increases in extreme precipitation are projected for many terrestrial regions by climate models in the Coupled Model Intercomparison Project (CMIP5) under different climate change scenarios (Kharin et al., 2013; Scoccimarro et al., 2013; Wang et al., 2017; Donat et al., 2019). However, at regional scale, the changes are uncertain and vary from region to region (O'Gorman, 2012; Kharin et al., 2013; Pfahl et al., 2017).

Observations indicate that the frequency and intensity of extreme daily precipitation events have already increased over large parts of the continents during the 20[th] century (Asadieh and Krakauer, 2015; Donat et al., 2016, 2019; Myhre et al., 2019; Papalexiou and Montanari, 2019). These changes are coherently captured by many CMIP5 models (Donat et al., 2016, 2019), though overall, significant inter-model spread is found in the historical period and even more in climate projections (Scoccimarro et al., 2013; Donat et al., 2016; Wang et al., 2017; Donat et al., 2019). Additionally, the models show some
bias in the magnitude of extreme daily precipitation (Stephens et al., 2010; Kopparla et al., 2013). The discrepancies between observations and climate models are potentially associated with the model-dependent sensitivity of extreme precipitation to the increase in temperature (Donat et al., 2016) and the model parameterization of subgrid scale physical processes which are relevant for extreme precipitation (e.g., Champion et al., 2011; Van Haren et al., 2015; Scher et al., 2017).

Many studies focus on the understanding of the nature of extreme daily precipitation in the historic period covering the last 50 to 100 years and their changes until the end of the 21$^{st}$ century under climate change scenarios (e.g., Asadieh and Krakauer, 2015; Donat et al., 2016, 2019; Wang et al., 2017; Myhre et al., 2019). Studying extreme events in the historical period is a reasonable approach as the analysis of such events with short duration requires continuous records of daily precipitation and such records are only provided by the modern instrumental observations and model simulations. Nevertheless, to understand the entire nature of these complex and rare events, it would be ideal to investigate their characteristics and long-term variability in the past when the anthropogenic influences on the climate were not present yet.

Reconstructions of past climate events based on natural proxies and historical documents can provide a glimpse of the nature of extreme precipitation and floods in the past (e.g., Brázdil et al., 2012; Kjeldsen et al., 2014; Machado et al., 2015; Steinschneider et al., 2016; Zheng et al., 2018). For instance, lake sediment records indicate that the frequency of floods has increased during the Little Ice Age in the Mediterranean French Alpine region (Wilhelm et al., 2012). This period is also coherent with increased flood occurrences in central Europe described in the 500-year documentary records on river catchments (Glaser et al., 2010). Frequent floods in this period seem to be associated with large-scale atmospheric circulation patterns such as the Atlantic low and Russian high (Jacobeit et al., 2003). In the semi-arid regions in the western United States, the frequency of extreme precipitation is inferred from tree ring-based reconstructions of the summer Palmer Drought Severity Index for the last 500 years (Steinschneider et al., 2016). The reconstruction indicates that the regions present a low-frequency variability of extreme precipitation that ranges over ample frequency bands, showing a variability from 2 to 30 years. Natural proxies and historical reconstructions are undoubtedly invaluable sources of information on extreme events for periods prior to the start of the modern instrumental era. However, they also pose some limitations: they are not continuous in time and scarcely distributed across the globe, therefore, providing more local aspects of such events.

Nowadays, earth system models that describe the physical processes within the climate system provide abundant and continuous data of global atmospheric variables, not only for the historical and future periods but also for the past (Schurer et al., 2013; Jungclaus et al., 2017; PAGES Hydro2k Consortium, 2017). As these state-of-the-art climate models are already utilized to understand the present mechanisms and future changes of extreme daily precipitation, they can also help to explore the past natural variability of extreme precipitation, assuming that the corresponding transient external forcing is properly adapted in simulations for the past (Schmidt et al., 2011; Jungclaus et al., 2017).

The purpose of this study is to analyze and characterize extreme daily precipitation for the period 1501 BCE to 1849 CE (before the industrial revolution) using the Community Earth System Model version 1.2.2. We base our analysis on simulations, which are either driven by orbital forcing alone or by all external forcing functions, including the new continuous volcanic record (Sigl et al., 2021). These simulations allow to distinguish between signals of orbital and volcanic forcing and internal variability. For the analysis, we employ the peak-over-threshold extreme value analysis (Coles et al., 2001) to characterize the 3351 year pre-industrial extreme daily precipitation on a global scale, to assess its long-term connection to externally forced variability, such as the changes in radiation driven by the orbital parameters and volcanic eruptions, to internal variability associated with large-scale circulation patterns, and to surface air temperature. The peak-over-threshold analysis is an amply used method to understand the characteristics of extreme events. For instance, the method has been used to study the characteristics

and recurrence periods of wind storm related variables (Della-Marta et al., 2009; Blender et al., 2017) and extreme precipitation
(Sugahara et al., 2009; Thiombiano et al., 2017) in different locations, and the association of extreme events with large-scale
modes of variability (Silva et al., 2016; Blender et al., 2017; Thiombiano et al., 2017).

This paper is organized as follows: in Sect. 2, we introduce the Community Earth System Model, the set-up of the model
simulations, and the observational data sets used to evaluate the model's ability to represent daily precipitation. Section 3
outlines the theoretical basis of the peak-over-threshold extreme value analysis and the methods employed for our analysis.
The results are presented in Sect. 4 where we first evaluate the climate model against reanalysis data, then show the distribution
of extreme daily precipitation globally, and finally, assess how the extreme daily precipitation is statistically related to orbital
forcing changes, internal modes of variability, and volcanic forcing. Lastly, conclusive remarks are presented in Sect. 5.

## 2   Data

### 2.1   Description of the model, simulations, and observational data

The Community Earth System Model (CESM; Hurrell et al., 2013) version 1.2.2 is a fully coupled general circulation model
that is composed of several component models: the Community Atmosphere Model version 5 (CAM5; Neale et al., 2010) for
the atmosphere, the Community Land Model version 4 (CLM4; Lawrence et al., 2011) for the land which includes a prognostic
carbon-nitrogen cycle, the Parallel Ocean Program version 2 (POP2; Smith et al., 2010) for the ocean, and the Community Ice
Code version 4 (CICE4; Hunke et al., 2010) for the sea ice. The spatial resolutions of the simulations are $1.9° \times 2.5°$ for
the atmosphere and land, and $1° \times 1°$ for the ocean and sea ice. The atmosphere is resolved at 30 and the land at 15 vertical
levels. The ocean has 60 vertical levels. The time resolutions of the simulations are 6-hourly and monthly, and the 6-hourly
precipitation is aggregated to a daily time resolution.

Using CESM, we perform two simulations covering the period 1501 BCE – 2008 CE (Table 1). The two simulations are
branched off from the last year of a spin-up simulation with perpetual forcings. The spin-up simulation is performed with
the orbital parameters and forcings set constant at 1501 BCE conditions, with a $CO_2$ level of 274.21 ppm (Bereiter et al.,
2015), a $CH_4$ level of 572.88 ppb, a $N_2O$ level of 262.79 ppb (Joos and Spahni, 2008), and a total solar irradiance (TSI) of
1360.38 $Wm^{-2}$ (Vieira et al., 2011; Usoskin et al., 2014). The land use and land-use changes (LULUC) are set to conditions
reconstructed for the year 850 CE (Pongratz et al., 2008). The spin-up simulation is run for 1405 years until it reaches an
equilibrium state.
The first simulation that branches off the spin-up simulation includes orbital parameters that vary in time from 1501 BCE to
2008 CE. All other forcings are kept the same as in the spin-up period (Table 1). Hence, this simulation includes only the effect
of the changes in the orbital parameters and internal climate variability. Hereinafter, this simulation is called the orbital-only
simulation.

The second simulation is run with all time-varying external forcings from 1501 BCE to 2008 CE (Fig. 1). The variables
for the external forcings are the total solar irradiance (TSI), greenhouse gas concentrations (GHG), volcanic sulfate aerosols
(VOL), and LULUC. The TSI is reconstructed from $\Delta^{14}C$ (Usoskin et al., 2014, 2016; Vieira et al., 2011) and obtained from

the Paleoclimate Modelling Intercomparison Project Phase 4 (PMIP4) database (https://pmip4.lsce.ipsl.fr/). For GHG up to 1849 CE, the annual records of $CO_2$ is obtained from Bereiter et al. (2015), and of $CH_4$ and $N_2O$ from Joos and Spahni (2008). These annual time series are smoothed by cubic spline interpolation. After 1850 CE, the annual GHG is extended using the records from Meinshausen et al. (2017). LULUC up to 1500 CE is based on the reconstructions from Pongratz et al. (2008) and after this year, it is merged with the reconstruction from Hurtt et al. (2011). Prior to 850 CE, the LULUC is set constant to 850 CE values and it varies afterwards. This second simulation is denoted as the full-forcing simulation (see Table 1).

The record of volcanic sulfate aerosols from 1501 BCE to 1979 CE is obtained from Sigl et al. (2021), and afterward until 2008 CE from Carn et al. (2016). To incorporate this record into the transient simulation, we use the Easy Volcanic Aerosol Model version 1.2 (EVA; Toohey et al., 2016) to generate the temporal and latitudinal distribution of the volcanic aerosols. Then, two modifications are applied to this volcanic aerosol distribution to attain an atmospheric response for the 1991 Pinatubo eruption, which is similar to the respective response in Gao et al. (2008), the volcanic forcing used in Paleoclimate Modelling Intercomparison Project (PMIP) Last Millennium experiments (Schmidt et al., 2011; Lehner et al., 2015; Otto-Bliesner et al., 2016): First, the total volcanic aerosols are scaled by a factor of 1.49. By applying the scaling, more similar responses of atmospheric temperatures and incoming solar radiation to the eruptions in the here presented simulation and Gao et al. (2008) is ensured. A similar scaling approach was used by Zhong et al. (2018), but with a slightly higher scaling factor of 1.79, as they used the 1815 Tambora eruption as reference. Second, the timing of the maximum peaks of the eruptions is shifted by four months in time after the respective eruption following the approach in Gao et al. (2008). After this peak, the volcanic aerosols decay smoothly as estimated by the EVA model.

To evaluate the daily precipitation obtained by CESM, the precipitation of CESM is compared to that of ERA5, the latest reanalysis product of ECMWF, during the period 1979–2008 CE (Hersbach et al., 2020). ERA5 uses the 2016 version of the ECMWF numerical weather prediction model and the integrated forecasting system Cy41r2 data assimilation (Hersbach et al., 2020). From ERA, we use the total precipitation at a temporal resolution of an hour and at a spatial resolution of $0.75° \times 0.75°$. To be compared against CESM, the hourly precipitation of ERA5 is accumulated to daily precipitation sums and the spatial resolution is interpolated to the coarser grid resolution of $1.9° \times 2.5°$ to be consistent with the resolution of CESM.

## 3 Methods

We apply the peak-over-threshold (POT) analysis to four datasets of daily precipitation anomalies: the CESM orbital-only and full-forcing simulations for the past period 1501 BCE–1849 CE, the CESM full-forcing and the ERA5 reanalysis for the present period 1979–2008 CE. The daily precipitation anomalies are calculated by subtracting the multi-year daily means, i.e., the means of the entire 1501 BCE–1849 CE period for the CESM past simulations and 1979–2008 CE for the present period in CESM and ERA5, from each daily precipitation value.

For the 1501 BCE–1849 CE simulations, the 99[th] percentiles relative to their distributions of the entire period are taken as a threshold to define extreme precipitation. For evaluation in the period 1979–2008 CE, the 95[th] percentiles are selected as a threshold in order to obtain enough extreme events during this rather short period.

## 3.1 Peak-over-threshold extreme value analysis

The POT approach (Coles et al., 2001) states that the values $y$ of a sequence of an independent random variable $x$, in our case, the precipitation anomalies, that exceed a certain threshold $u$, i.e., the 99th or 95th percentiles, are asymptotically distributed following a generalized Pareto distribution (GPD) with the density distribution function given as:

$$H(y) = \begin{cases} 1 - (1 + \frac{\xi y}{\sigma})^{-1/\xi} & \text{for} \quad \xi \neq 0 \\ 1 - \exp(-\frac{y}{\sigma}) & \text{for} \quad \xi = 0 \end{cases} \tag{1}$$

where $y = x - u$ are the positive exceedances of daily precipitation anomalies, $\sigma$ is the scale parameter that characterizes the spread of the distribution and the scaling of the exceedances $y$, and $\xi$ is the shape parameter that represents the upper-bound and tail behavior of the distribution (Sugahara et al., 2009; Blender et al., 2017). When $\xi > 0$, the upper-bound is infinite and the distribution has a heavy tail. For $\xi = 0$, the upper-bound is also infinite, but the tail shape is lighter as the distribution decays exponentially. When $\xi < 0$, the distribution has a finite upper-bound $y \leq -\sigma/\xi$ (above this upper-bound the probability vanishes) and a thin tail (Coles et al., 2001). When the number of exceedances is small and the estimated $\xi$ is negative, there is a bias in the estimation of $\xi$ towards a larger standard error (Blender et al., 2017). This occurs because since any sample has a finite maximum, there is a bias towards estimated distributions with an upper limit, hence the negative estimated shape parameter (Giles et al., 2016). The behaviors of the density distribution with different values of scale and shape are illustrated in Fig. 2.

Equation (1) describes the stationary GPD model, in which the scale and shape parameters remain constant. From the stationary GPD model, the $T$-year return level $y_T$ associated with the return period $T$ (Coles et al., 2001; Khaliq et al., 2006) can be estimated as:

$$y_T = \sigma[\zeta_u T^\xi - 1]/\xi \tag{2}$$

where $\zeta_u = P(x > u)$ is the ratio of exceedances in the sample. For the 95th percentile this is $\zeta_u = 0.05$, and for the 99th percentile this is $\zeta_u = 0.01$.

The POT analysis requires independent values among exceedances (Coles et al., 2001). Hence, the record of extremes needs to be de-clustered to reduce the persistence among the clustered extremes. For this, we de-cluster the extreme precipitation at each grid point by taking the maximum value within each cluster. Each cluster is composed of consecutive days of extremes and the extremes separated by a maximum of one day to other extremes. In other words, the minimum temporal distance allowed between the extremes within a cluster is one day, and between the clusters is two days (Coles et al., 2001). The result of the de-clustering can be quantified through an extremal index, which is the ratio between the number of extremes after being de-clustered and the initial number of extremes. These de-clustered exceedances are the values used for the analysis.

Then, a GPD is fit to the de-clustered extreme daily precipitation anomalies to generate a stationary GPD model with a specific scale and shape parameter at each grid point. The estimation of scale and shape parameters is performed using the maximum likelihood method (Coles et al., 2001; Sugahara et al., 2009).

The parameter estimation for a stationary GPD model through a maximum likelihood is given as follows: under the assumption that the exceedances $z_1,...,z_k$ are independent variables where k is the number of exceedances, the log-likelihood function $l$ for the parameters $\sigma$ and $\xi$ is:

$$\begin{cases} l(\sigma,\xi) = -k\log(\sigma) - (1+\frac{1}{\xi})\sum_{i=1}^{k}\log(1+\frac{\xi z_i}{\sigma}) & \text{for} \quad (1+\sigma^{-1}\xi z_i) > 0 \quad, \quad i=1,...,k \\ l(\sigma) = -k\log(\sigma) - \sigma^{-1}\sum_{i=1}^{k}z_i & \text{for} \quad \xi = 0 \end{cases} \tag{3}$$


Having the parameter vector $\beta$ with $\beta = [\sigma,\xi]$, the maximization of the pair of log-likelihood $l(\sigma,\xi)$ with respect to the $\beta$ is performed. This maximization leads to the maximum likelihood estimate of the scale $\sigma$ and shape $\xi$. The maximization is done numerically, as no analytical solution is possible (Coles et al., 2001).

In the case of non-stationary GPD models (Coles et al., 2001; Blender et al., 2017), the scale and/or shape parameters vary 200 linearly with time $t$, or with other external variables (covariates). In our analysis, we only allow scale parameters to change but shape parameters remain constant, similar to the approach used by Sugahara et al. (2009) and Blender et al. (2017). In our non-stationary GPD models, the scale parameter in Eq. (1) becomes

$$\sigma(t) = \sigma_0 + \sigma_1 t \quad \text{or} \quad \sigma(t) = \sigma_0 + \sigma_1 C(t) \tag{4}$$

with $C(t)$ being the time series of a covariate. Again GPDs are fit to the de-clustered extreme daily precipitation anomalies 205 to generate non-stationary time- and covariate-GPD models. Time dependent scale and shape parameters for the non-stationary model are also estimated using the maximum likelihood method following Eq. 3, assuming $\sigma(t) = \sigma_0 + \sigma_1 C(t)$ or $\sigma(t) = \sigma_0 + \sigma_1 t$ (Coles et al., 2001; Sugahara et al., 2009). The parameter vector $\beta$ in this case is $\beta = [\sigma_0,\sigma_1,\xi]$ (El Adlouni et al., 2007).

The performance of the non-stationary GPD model is measured relative to the stationary GPD model using the deviance statistics (Coles et al., 2001; Blender et al., 2017):

$$D = 2(L_1 - L_0) \tag{5}$$

where $L_1$ and $L_0$ are the log-likelihoods of the non-stationary and the stationary GPD model, respectively. The deviance statistics follows the $\chi_m^2$ distribution, and with $m = 1$ and a 99% confidence interval, a threshold for $D$ is 6.634. Hence, the relative performance of the non-stationary GPD model is statistically significant at a 99% confidence interval when $D$ is larger than 6.634. This indicates that the non-stationary GPD model is significantly better than the stationary GPD model at 99% 215 confidence interval, and the non-stationary GPD explains the variability of extreme precipitation at a grid point better than the stationary GPD. If this is the case, we assume that there is an association between the corresponding covariate, or the time variable, with extreme precipitation at that grid point or region.

### 3.2 Non-stationary GPD models with external forcings, modes of internal variability and surface air temperature

We use several external forcings, modes of internal variability, and surface air temperature anomalies (TS) as covariates of the 220 GPD models for the 1501 BCE–1849 CE simulations. For the external forcings, we consider five variables in total: three orbital

parameters combined in one variable (ORB; eccentricity (ECC), longitude of perihelion (PER) and obliquity (OBL); Berger, 1978), TSI, insolation (INS), clear sky net surface short wave radiation (FSN) and VOL (Fig. 3). ORB and TSI are annually resolved one-dimensional time series. INS and FSN are the output variables from the model simulations and resolved monthly and spatially at each grid point. The VOL forcing is monthly and latitudinally resolved and is already described in Sect. 2.1.

INS, FSN, and VOL are annually-averaged to obtain a yearly resolution. This procedure is applied to have consistent time resolutions among all external forcing variables and to exclude the effects of seasonality. Finally, all variables are normalized with respect to their 3351 year means and interpolated to a daily time resolution.

INS and FSN can be interpreted as the variables that reflect the combined effects of ORB and TSI on regional and global scales. Additionally, FSN also includes the effects of volcanic eruptions by exhibiting negative peaks on its time series after

some strong eruptions, but only in the transient simulation (Fig. 3).

Each of these variables is included in the scale parameters of each individual GPD model in Eq. 1 using Eq. 4, except for ORB. For ORB, all three parameters are incorporated together in the scale parameter of one GPD model as:

$$\sigma(t) = \sigma_0 + \sigma_1 ECC(t) + \sigma_2 OBL(t) + \sigma_3 PER(t) \tag{6}$$

The resulting combined effect is that $\sigma$ increases approximately linear with time $t$.

For the volcanic forcing, in addition to the analysis of the entire period, we also assess the short-term influence of volcanic eruptions on extreme precipitation. For this, we select a three-year period before (pre-eruption period) and a three-year period after (post-eruption period) all eruptions from the 1501 BCE – 1849 CE transient full-forcing simulation. Only the eruptions that fulfill the following conditions are included in the analysis: tropical eruptions that occur in January and exceed 2.66 Tg of volcanic stratospheric sulfur injection (VSSI). This reference VSSI is based on the El Chichon eruption in 1982 CE, whose

effects on the radiation and climate were clearly detectable (Hofmann, 1987). We also assure that no other eruption has occurred 5 years before and after each of the selected eruptions. After applying these criteria, the total number of eruptions included in the analysis is 57. These eruptions occur in the same month (January) and at similar latitudinal locations (tropical). Therefore, the asymmetric cooling due to extratropical eruptions (Oman et al., 2005; Schneider et al., 2009) and season-dependent climatic responses to tropical eruptions (Stevenson et al., 2017) are not considered for the analysis.

It is important to mention that all reconstructed external forcings such as TSI and volcanic eruptions contain inherent uncertainties derived from reconstruction models or methods and from the dating of events (Sigl et al., 2015; Jungclaus et al., 2017; Matthes et al., 2017). An attempt to reduce such uncertainties is an active research topic (e.g., Sigl et al., 2015; Matthes et al., 2017) that is beyond the scope of this study. A possible implication of uncertainties from the external forcings in our analysis is briefly discussed in the result section (Sect. 4.4).

A stationary GPD fit (Eq. 1) is applied to the pre-eruption and post-eruption periods separately to estimate the return periods of extreme precipitation, then to assess the post-eruption changes in extreme precipitation. Additionally, the numbers of days and the intensities of extremes for these two periods are calculated. As strong volcanic eruptions influence the evolution of ENSO states (McGregor and Timmermann, 2011; Ohba et al., 2013; Wang et al., 2018; McGregor et al., 2020), we also identify imprints of ENSO on post-eruption extreme precipitation by splitting the years of eruptions (year 0) into three ENSO

states, based on the Niño3.4 index: El Niño (> 0.5 K), La Niña (< -0.5 K) and neutral state (> -0.5 K and < 0.5 K). Then, the numbers of extremes in each ENSO state are counted and compared to the numbers in the same states during the year before the tropical eruptions.

To generate the non-stationary GPD models with the internal modes of variability as covariates, we use eight modes of variability: Eastern Atlantic–West Russian Pattern (EA-WR), North Atlantic Oscillation (NAO), Northern Annular Mode (NAM),
Pacific Decadal Oscillation (PDO), Pacific North American (PNA) pattern, ENSO, Southern Annular Mode (SAM) and Pacific South American 1 (PSA) mode. The procedures to calculate the modes of variability are explained in the supplement. These modes of variability are both captured by CESM and ERA5, as the patterns are comparably similar among the data sets (Figs. S1, S2, and S3 in the supplement). Then, we constrain the regions of action of some of these modes to only one hemisphere: EA-WR, NAO, NAM, PDO, and PNA are only associated with extreme precipitation in the Northern Hemisphere, while SAM
and PSA only influence extreme precipitation in the Southern Hemisphere. There is no spatial restriction for ENSO; therefore, ENSO can be associated with extreme precipitation in both hemispheres.

All of these modes have a monthly time resolution and each of them is inserted in a GDP model without an interpolation to the daily time resolution. For the internal variability, we assume that what influences the extreme daily precipitation is not the daily fluctuations of these modes but their monthly mean values.
TS is obtained by subtracting the 1501 BCE–1849 CE monthly means of surface air temperature from each monthly value in the simulations. Two kinds of TSs are considered for the non-stationary analysis: one is the globally-averaged means of TS (TS-G; Fig. S4) and another is the spatially (latitude and longitude) gridded TS (TS-R). The former is to assess the influence of global temperature and the latter is to assess the influence of regional temperature on daily extreme precipitation. Both TSs are resolved monthly and similar to the modes of internal variability, they are not interpolated to daily time resolution. The
influences of TSs are compared against to those of the modes of variability.

### 3.3 Statistical tests used for the model evaluation against ERA5

We compare the daily precipitation in CESM and ERA5 for the present period 1979–2008 CE in order to evaluate the model's ability to depict the daily precipitation. For this, we analyze the 30-year global and land averaged trends of the annual mean daily precipitation and the spatial means of the entire and extreme (the values above the 95[th] percentiles) daily precipitation.
The signs of the monotonic trends of the global and land averaged total daily precipitation are compared to each other and their statistical significance is calculated using the non-parametric Mann–Kendall trend (M-K) test (Mann, 1945; Wilks, 2011; e.g., Westra et al., 2013). Under the null hypothesis, the M-K test assumes no trend in the time series. The means of the total and extreme daily precipitation between CESM and ERA5 are compared using the non-parametric Mann–Whitney (M-W) U-test (Mann and Whitney, 1947; Wilks, 2011; e.g., Kim and Raible, 2021) at each grid point. The null hypothesis of the M-W
test states an equal distribution of the two data sets. To assess the similarity of the spatial patterns, the Pearson $r$ correlation coefficients are calculated using the spatial mean values of total and extreme precipitation between CESM and ERA5. The null hypothesis of the test assumes no correlation between two data sets. Hence, if the null hypothesis is accepted, no significantly similar spatial precipitation pattern between both datasets is observed.

## 4 Results

 ### 4.1 Comparison between ERA5 and CESM in the present period 1979–2008 CE

In this section, we compare the daily precipitation in ERA5 and CESM for the period 1979–2008 CE to evaluate the model's ability to represent the mean and extreme daily precipitation. For this comparison, we use the full-forcing simulation.

The 30-year global and land-only averaged annual means of the daily precipitation anomalies in ERA5 and CESM are shown in Fig. 4. During 1979–2008 CE, both ERA5 and CESM show a small but statistically significant positive trend for the global daily precipitation at a 99% confidence interval (Fig. 4a). However, when only the daily precipitation anomalies over land are considered, significant positive trends are absent in both data sets (Fig. 4b). ERA5 shows no statistically significant trend of the daily precipitation over land, while CESM indicates a slight negative but significant trend during this 30-year period. This difference in trends between the global and land averaged daily precipitation suggests that the changes in the mean daily precipitation over land during the last few decades are more heterogeneous than the global changes of daily precipitation resembling findings of Contractor et al. (2021).

Regarding the spatial distribution, some differences between ERA5 and CESM are evident, particularly in terms of the magnitudes of the mean daily precipitation (Fig. 5a). Compared to ERA5, CESM largely underestimates the daily precipitation over the tropical and North Pacific Oceans, India, central Asia, Australia, southern South America, almost all of Africa, and western North America. The model overestimates the daily precipitation over the tropical Atlantic, northern and central South America, and large parts of Europe among others. The differences in the magnitudes are also present in the extreme daily precipitation (daily precipitation above the 95 [th] percentiles relative to their 1979–2008 CE distributions) (Fig. 5b). However, CESM overestimates extreme daily precipitation in regions where the mean daily precipitation is underestimated and vice versa. The discrepancy between the reanalysis or the observation and the model simulation in the magnitudes of the mean and extreme daily precipitation is a known problem in many climate models (Stephens et al., 2010; Flato et al., 2014). This discrepancy demonstrates again the difficulties of models to realistically represent physical processes related to precipitation, mainly associated with short-lived and spatially small scale events (e.g., Champion et al., 2011; Van Haren et al., 2015; Scher et al., 2017).

Regardless of these differences in the magnitudes of the mean and extreme daily precipitation, CESM represents relatively well the spatial patterns of mean daily precipitation. CESM distinguishes properly the drier from wetter regions, which are identified as the regions with precipitation values below and above the spatial 80[th] percentile of the total mean precipitation, respectively (brown contour line in Fig. 5). The spatial Pearson correlation coefficients of the mean total and extreme precipitation between ERA5 and CESM are 0.88 and 0.92, respectively, and both values are statistically significant at the 99% confidence interval. These values demonstrate that the model is able to realistically represent the mean total and extreme spatial pattern of daily precipitation.

The POT analysis is applied to the extremes in ERA5 and CESM, and estimated parameters of the stationary GPD models from the analysis are presented in Fig. 6. The extremal indices, which are the ratios between the numbers of de-clustered extremes and initial extremes, show reduced numbers of extremes over the tropical oceans after de-clustering. In some regions

in the tropics, the de-clustering method leaves only around 40% of the initial numbers of extremes, indicating that the reduction of extreme events is particularly strong over this latitudinal belt, a known region of convective organization with temporal

clustering. Over the extratropics, the extremal indices range from 0.8 to 1, meaning that de-clustering does not strongly affect the number of extremes in these regions. The extremal indices illustrate that over the tropics, mainly over the oceans, clustered precipitation events that last for several days are common, while short-lived extreme precipitation events are prevalent over the extratropics. Hence, de-clustering causes a strong reduction in the number of extremes in the tropics, which is more pronounced in the CESM simulation.

The scale parameters in ERA5 and CESM, which indicate the spread of extreme precipitation, largely follow the spatial pattern of the mean extreme daily precipitation in Fig. 5b. The parameters show higher values over wetter regions and comparably lower values over other regions. This behavior of the scale parameters is expected as the parameter is related to the scaling of the exceedances in the density distribution, which is associated with the mean values of the extremes in Fig. 5b (see also Fig. 2). Hence, the difference in the scale parameters between ERA5 and CESM is expected, as both data sets present different

mean extreme precipitation.

For the shape parameter, CESM shows more regions with negative shape parameters than ERA5 and more standard errors from the parameter estimation over these regions. As mentioned in Sect. 3.1, this occurs due to the small sample size for the parameter estimation over the regions where the shape parameters are negative (Blender et al., 2017). Excluding these regions, ERA5 and CESM share some positive values over the same regions and some coherent regions with relatively high shape

parameters in the Indian Ocean, the southern Atlantic, and Pacific Oceans.

The return periods of extreme precipitation in both CESM and ERA5 largely follow the pattern of the extremal indices. It exhibits long return periods over the tropics where the extremal indices are lower and short return periods over the extratropics where the extremal indices are higher. This behavior of the return periods is expected, as a low number of extremes indicates a low occurrence of events, therefore, an increased return period of the events. Note that the return period over the tropics should

be interpreted as a return period of a clustered event instead of a return period of a single or a short-lived precipitation event.

In summary, some differences between ERA5 and CESM exist in the parameters of the GPD models due to differences in the mean extreme daily precipitation. Still, the spatial patterns of the GPD parameters and return periods of simulated extreme daily precipitation anomalies resemble the patterns of the reanalysis, presenting coherent regions with maximum and minimum parameter values. Wet and dry regions and the regions with maximum and minimum GPD parameters are in general

coherent between ERA5 and CESM. This indicates that the model represents the large-scale spatial patterns of the mean and extreme precipitation relatively well. The focus of this study is on the long-term changes and characteristics of extreme daily precipitation on a global scale rather than on the characteristics or impacts of a handful single events. Therefore, we use the daily precipitation from CESM as it is in the next section, i.e., without applying any further corrections, such as bias correction (e.g., Chen et al., 2020) or downscaling methods (e.g., Yang et al., 2012). Still, we take into account the differences between

ERA5 and CESM in the magnitudes of extreme precipitation when interpreting our results and discussing possible implications in the conclusions.

## 4.2 Distribution of extreme daily precipitation for the period 1501 BCE–1849 CE in CESM

Here, the POT analysis is applied to the time series of extreme daily precipitation in the 3351 year orbital-only and full-forcing transient simulations to generate the stationary GPD models at each grid point and to illustrate the characteristics of the distribution of extreme precipitation. The thresholds for the extremes (the 99[th] percentiles of daily precipitation relative to their distributions of the entire period), the means of all precipitation above the extreme percentile limit, and extremal indices are presented in Fig. 7. The parameters of the stationary GPD models from the full-forcing simulation are shown in Fig. 8. The POT analysis of the orbital-only simulation exhibits similar results; thus, the corresponding figures are not shown here.

The spatial patterns of the mean extreme precipitation and extreme indices (Fig. 7) are similar to those during the present period in Fig. 5b, discerning the wetter and drier regions. Similar to results found in Fig. 6, the scale parameters (Fig. 8a), which describe the spread of the distribution or scaling of the extremes, largely follow the spatial pattern of the mean extreme precipitation with higher values over the wetter regions in the Pacific and Indian Oceans, south and east Asia, east and west coasts of North America among others. Relatively lower values of the scale parameter are found in Europe, Africa, and northern South America.

For the shape parameters (Fig. 8b), the regions with higher estimation errors (striped area) are reduced compared to Fig. 6. This is expected because the numbers of extreme precipitation events increase so that the parameter estimation is more reliable. Positive shape parameters dominate over the tropical and west Pacific and Indian Oceans. Over land, the wetter regions in Asia, the southeast coast of South America, and the east and west coasts of North America dominate more negative values. Similar to Fig. 5, the spatial pattern of the return periods (Fig. 8c) exhibit high return periods located over the tropics and relatively lower values over the extratropics.

The scale and shape parameters together (Fig. 8d) describe the generalized Pareto distributions that characterize the density distributions of extreme precipitation at each grid point. The density distribution of extreme precipitation is separated into four types given in Fig. 2: I) high scale - positive shape characterized by a higher spread, heavy tail and no-upper bound of extremes ( Fig. 2a); II) high scale - negative shape characterized by a high spread, thin tail and an upper-bound (Fig. 2b); III) low scale - positive shape characterized by a low spread, heavy tail and no upper-bound (Fig. 2c); and finally, IV) a low scale - negative shape is characterized by a low spread, thin tail and an upper-bound of extremes (Fig. 2d). High and low values of scale parameters are defined as the values above and below the median of all scale parameters in Fig. 8a. In Fig. 8d, clustering of the same types of distribution is noticeable. For instance, the type I (high scale - positive shape) dominate over the west and the tropical Pacific Ocean; the type II (high scale - negative shape) are located over the south and north Pacific and Atlantic Oceans, south and east Asia, and eastern North America; the type III (low scale - positive shape) are found over some regions in the Southern Ocean, northern Asia, Amazon, and Africa; lastly, the type IV (low scale - negative shape) distributions occur over some regions in the Southern Ocean and in the northernmost Atlantic Ocean. In general, over the ocean, the types of distribution seem to be more homogeneous than those over land. In addition, there are clear differences in the types of distribution between the Pacific and Atlantic sectors and between the northern high and southern low latitudes. To conclude,

the results show that the POT analysis is able to identify the large-scale characteristics of extreme precipitation by separating distinct and coherent regions for similar types of distributions.

## 4.3 Association of extreme precipitation with external forcings including the tropical volcanic eruptions

To assess the long-term influence of externally forced variability on extreme precipitation, the time series of the variables mentioned in Sect. 3.2 (Fig. 3, Figs. S1 and S2) are included as covariates in the non-stationary GPD models (Eq. 4). Then, the deviance statistics $D$ (Eq. 5) are calculated to quantify the performance of the covariate GPD models in the orbital-only and full-forcing simulations relative to the stationary GPD models in Fig. 8.

The regions, where each covariate, the time and external forcing GPD models, outperform the stationary GPD models with the statistically significant $D$ values (with $D$ more than 6.634 at 99% confidence interval) is shown in the supplement Fig. S5 and S6 for each of the simulations. The regions where both orbital-only and full-forcing simulations share a common significant $D$ are presented in Fig. 9. We assume that these regions with coherent $D$ between the two simulations are where the extreme precipitation is truly influenced by the covariates.

The time, ORB, and INS share some common regions over the tropical Atlantic, the Indian Ocean, central Asia, and central Africa. These variables present a time series with a monotonic linear trend. Thus, over these regions, there is a linear increase in extreme precipitation during 1501 BCE–1849 CE. The influence of TSI is spread randomly across the globe, implying no robust association of this variable with the variability of extreme precipitation. FSN, which considers the overall changes in the net surface shortwave radiation, exhibits similar regions of association with extreme precipitation to those in ORB and INS, mostly over the tropics. Still, the regions of association of FSN with extreme precipitation are larger, as additional internal effects involved in the surface radiation balance and not only the external forcing are included in this covariate. For example, the regional temperature variability affected by internal variability has an impact on FSN. Overall, the roles of the external forcing in the variability of extreme precipitation during 1501 BCE–1849 CE are constrained to certain small regions shown in Fig. 9.

The VOL shows a random spread similar to TSI. This relatively limited influence of volcanic aerosols on extreme precipitation can be related to the time scale of the influence of volcanic eruptions on the climate. It is known that the effect of volcanic eruptions on precipitation is only in the short-term, in the range of a few years (Robock, 2000; Iles et al., 2013; Iles and Hegerl, 2014). Hence, volcanic eruptions may not alter the long-term variability of extreme precipitation analyzed here. Considering the time scale of the effects of volcanic eruptions on precipitation, short-term influences of volcanic eruptions on extreme precipitation are specifically analyzed by taking the periods of three years before and three years after the eruptions (Sect. 3.2).

The differences in mean, return period, and number of extreme precipitation between the post- and pre-eruption periods are shown in Fig. 10. In terms of the intensity of extreme precipitation (Fig. 10a), the tropical equatorial Pacific region presents a statistically significant increase after eruptions. Other regions with significant increases or decreases are spread across the globe. It is noted that the post-eruption changes in the intensity of extreme precipitation are not as evident as the changes in the number of extreme precipitation (Fig. 10b and c). The return periods of extremes in Fig. 10b indicate a decrease after eruptions

in the tropics, west Indian Ocean, and lands in the mid-latitudes, including the southern US, central South America, and central and southern Europe. An increase is found in Asia, Australia, Africa, northern Europe, northern South America, and northern North America. These changes in the return periods reflect the changes in the number of extreme precipitation (Fig. 10c): where the return periods decrease, the number of extreme daily precipitation increases, and vice versa. Moreover, the percentages of changes in the number of extremes are clearly higher than those in the intensity of extremes. This result indicates that except for the tropical ocean and some small isolated regions, tropical eruptions influence more clearly the frequency than the intensity of extreme precipitation.

The patterns of the return period and the number of extremes in Fig. 10b and c resemble the precipitation anomaly pattern induced by El Niño events (Dai and Wigley, 2000), which is mainly characterized by an increase in precipitation over the tropical equatorial Pacific Ocean and the tropical regions from America to Africa, and a decrease over the eastern Pacific sector including Australia and Indonesia. Although, there is a lack of consistency among natural climate proxies in the responses of ENSO to tropical volcanic eruptions during the last millennium (Dee et al., 2020), model-based studies largely demonstrated that tropical volcanic eruptions alter the states of ENSO and can lead to El Niño-like conditions after the peaks of eruptions (McGregor and Timmermann, 2011; Wang et al., 2018; McGregor et al., 2020). Moreover, precipitation responses to volcanic eruptions are magnified during El Niño state after eruptions (Stevenson et al., 2016).

Fig. 10d exhibits the differences in the number of extreme precipitation during each of the ENSO states between the year of eruption (year 0) and the year before the eruption. Increases in the number of extremes are noticeable over a large part of the globe during El Niño phases in year 0, also over regions where the number of extremes increases in Fig. 10c. During La Niña, the anomalies are opposite to those during El Niño, presenting negative values over the tropical Equatorial Pacific, central South America, and southern Africa. During the neutral state, an increase in the frequency is evident over the tropical equatorial Pacific region, same as during El Niño phases, while in other regions, decreases in the frequency are found. All ENSO states show a reduction in the number of extremes over the Asian monsoon region in year 0. The result indicates that the increases and decreases in the frequency of extremes over large parts of the globe in Fig. 10a and b are explained by different ENSO states, mostly by El Niño and the neutral condition, influenced by tropical eruptions. Besides, the decreases in extremes in the high-latitudes are also connected with the cooling caused by the volcanic eruptions, as extreme precipitation depends on the change of the atmospheric moisture availability, which is modulated by the atmospheric temperature (Pall et al., 2007; Myhre et al., 2019).

## 4.4 Association of extreme precipitation with large-scale circulation patterns and surface air temperature

The same procedure as in the previous section (Sect. 4.3) is repeated to find the association between the extreme precipitation and the modes of internal variability. At each grid point, the $D$ statistics which are higher than 6.634 (Eq. 5) of all modes of variability are compared among each other, and the one with the maximum $D$ statistics is selected. Hence, the mode of variability with the maximum $D$ statistics is the variable that explains best the occurrence of extreme precipitation compared to other modes of variability and stationary conditions.

Fig. 11a shows the regions where the orbital-only and the full-forcing simulations coherently influence extreme precipitation through the same mode of variability. In general, the modes of internal variability GPD models outperform the stationary models across the globe. This result indicates that modes of variability play a more important role in explaining the long-term variability of extreme precipitation in the pre-industrial period than the natural external forcings do. In general, the modes of variability exhibit a significant connection to extreme precipitation in the vicinity to their regions of action. For instance, SAM is dominant in the Southern Ocean, and NAO and NAM in the North Atlantic region. The influence of ENSO is broader. Over land, ENSO dominates most of the Southern Hemisphere, eastern North America, southern Asia, and the eastern Mediterranean region. Although the connection between ENSO and extreme precipitation is strong and important (Kenyon and Hegerl, 2010), we also assume that this dominance of ENSO is partially related to the bias in CESM towards an overestimation of ENSO amplitudes (Stevenson et al., 2018). In Europe, the roles of EA-WR and PNA dominate over the western regions and NAM and NAO in the northern regions. In North America, PDO is the leading mode in the western regions and ENSO in the southeastern and central regions. Small areas are influenced by NAM and PNA. In the Southern Hemisphere, PSA and ENSO are dominant over land, while a slight influence of SAM is found over north-western Australia.

The result does not vary much when TS-G is included as covariate (Fig. 11b). Only a few sporadic points that denotes the influence of TS-G on extreme precipitation appear over the Southern Hemisphere, indicating that the changes in the global mean temperature affect little the long-term variability of extreme precipitation compared to the modes of variability. However, the influence of the surface temperature in Fig.11b changes when the regional temperature anomalies TS-R instead of TS-G are included as covariates (Fig. 11c). The dominance of TS-R over the modes of variability is highlighted in many land areas and the tropical oceans. The land areas where TS-R is more important are found largely at the extratropical latitudes in the Northern and Southern Hemispheres, covering a large part of northern Asia and North America, southern South America and Africa, Australia, the African transition zone, and the Arabian Peninsula. It is clear that over northern Asia, the influences of PNA and NAM that appeared previously in Fig. 11a and b are masked by those of TS-R. TS-R also prevails over the tropical Pacific, where ENSO takes place. In this region, the values of TS-R overlap with the NIÑO index, which is calculated as an average of the surface temperature anomalies of the same area. Hence, it is reasonable to interpret the predominance of TS-R in the tropical Pacific to be similar to the influence of ENSO. However, this pattern of TS-R simply indicates that the regional temperatures are statistically more associated with the regional extreme precipitation than the averaged temperature condition, thus the NIÑO index, over the area.

The here found associations of TS-R with extreme precipitation over land are in line with the preceding studies (Pendergrass et al., 2015; Sillmann et al., 2017; Sun et al., 2021), which have demonstrated that the present-day and future extreme precipitation are regulated by surface temperature. Here, it is also shown that TS-R does not outperform all the modes of variability over the entire land areas. There are regions, including some in the extratropics, where the modes of variability still play more important roles in regional extreme precipitation. Some of these regions are North America, northern South America, west- and southern Europe, and southern Asia, where ENSO, PDO, EA-WR, and PSA exhibit statistically significant associations.

Overall, the result indicates that both the modes of variability and regional temperature are more important than external forcings in the long-term variability of extreme precipitation during the pre-industrial 3351 years. Although the linear increase

in extreme precipitation due to the externally forced variability is present in some regions, this influence is masked when internal variability and regional surface temperature are included. Moreover, this limited influence of external forcings on extreme precipitation signifies that the inherent uncertainties of external forcings have a minimal effect on the characterization of pre-industrial extreme precipitation.

## 5 Conclusions

We have examined the characteristics of pre-industrial extreme daily precipitation and its long-term changes and association to externally forced and internal variability during 1501 BCE–1849 CE. The period is of particular interest as the orbital parameters have progressively changed during the late Holocene (Wanner et al., 2008) and many civilizations had flourished and vanished during this period. Thereby, the role of changes in climate on these societal changes is also highly debated (e.g., Hodell et al., 1995; DeMenocal, 2001; Büntgen et al., 2011; McConnell et al., 2020). Our study is based on climate simulations from CESM1.2.2, which cover the period 1501 BCE–2008 CE, and the peak-over-threshold (POT) extreme value analysis is used to analyze extreme daily precipitation. The main findings of this study are the following:

First, regions with similar statistical distributions of extreme precipitation are identified using the POT analysis. We have distinguished the regions in four different density distributions of extreme precipitation, and these regions tend to cluster spatially. Clear differences in the distributions of extreme precipitation are observable between the Pacific and Atlantic sectors and between the northern-high and southern-low latitudes.

Second, past variability of extreme precipitation is strongly associated with large-scale modes of variability such as ENSO, NAM, PNA, EA-WR, and PSA, among others and regional surface temperature. Largely, the modes of variability present significant association with extreme precipitation in the vicinity to their regions of action. In this study, the regions where extreme precipitation is more associated either by a mode of variability or by the regional surface temperature are distinguished. Regional surface temperature is linked with extreme precipitation in general over lands at the extratropical latitudes and in the tropical oceans. In other regions, the influences of modes of variability are still dominant. Some limited regions show an association of extreme precipitation with changes in the insolation caused by changes in the orbital parameters during this period. This association is reflected by a linear increase in extreme precipitation. The POT analysis specifies geographical regions where the association with the climate variability is statistically significant, though it does not elucidate in which way the climate variability influences extreme precipitation. Understanding how each forcing and mode influences extreme precipitation is beyond the scope of this study, as additional dedicated sensitivity simulations and analysis would be required.

Lastly, changes in the frequency of extreme precipitation are remarkable after strong tropical eruptions. Significant changes after tropical eruptions occur in the return period. Hence, the frequency of extreme precipitation increases over the extratropical regions and the tropical Pacific and decreases in others. These post-eruption changes in the frequency of extreme precipitation are associated with the ENSO states, which are also influenced by the volcanic eruptions and agree with Stevenson et al. (2016). The influence of volcanic eruptions on extreme precipitation is only noticeable in the short term up to a few years (Iles et al.,

2013; Iles and Hegerl, 2014; Stevenson et al., 2016). Statistically significant changes in the intensity of extreme precipitation are noted but are more heterogeneously distributed across the globe.

It is important to mention that some caveats need to be clarified in this study. There is a clear discrepancy between ERA5 reanalysis and CESM in representing the intensity of extreme precipitation. This is a problem that still many earth system models suffer from (Flato et al., 2014; Wang et al., 2017), and active research to reduce this difference is currently undertaken (Kopparla et al., 2013; Shields et al., 2016; Kawai et al., 2019). The fact that some regions over- or underestimate extreme daily precipitation may have implications for our results. For instance, the parameters of the generated statistical models are higher or lower than the values from the reanalysis. These differences in the parameter values can affect the estimation of return periods of extreme precipitation. However, we noted that the mean spatial patterns are rather similar between the two data sets. Therefore, the general conclusion on the global-scale spatial distributions of extreme precipitation should not be affected. Another point is that this analysis is based on a single climate model, CESM; therefore, the result is strongly influenced by the model-dependent internal variability (Fasullo et al., 2020). For instance, strong ENSO amplitudes in CESM may overrate the influences of this mode of variability on the global and regional extreme precipitation.

Nevertheless, this study provides a new approach to examine the nature of extreme daily precipitation in the past: we showed that continuous long climate model simulations help to partially understand the variability of extreme daily precipitation in the pre-industrial period. More volcanic eruptions are included from the newly available long record (Sigl et al., 2021; Dallmeyer et al., 2021), which can increase the robustness of the analysis of post-eruption extreme daily precipitation. Moreover, it is noted that the POT analysis is useful to discern regions where extreme precipitation is influenced by different climate factors.

Nowadays, understanding the variability of extreme precipitation is important to illustrate the entire natural mechanism of these rare events and to predict their future changes better. In the current context where the information on these events in the pre-industrial past is limited, studies based on available long climate simulations, such as the one employed here, can be a valuable contribution to complement the information on extreme precipitation. More modeling studies on extreme events in the past are needed to certify the results presented. This approach would clearly help us to understand the nature of these capricious, rare, and intense events in more detail, to improve the model-dependent representation of such events, and to distinguish the roles of the external and internal variability on them.

*Code availability.* The peak-over-threshold extreme value analysis was performed using two R packages: $ismev$ (https://CRAN.R-project.org/package=ismev) and $extRemes$ (https://CRAN.R-project.org/package=extRemes; Gilleland and Katz, 2016). A python script to calculate and plot the maximum D statistics is stored on https://github.com/wmk21/EVT-D-statistics-comparison.

*Data availability.* NOAA Extended Reconstructed Sea Surface Temperature Version 5 (ERSSTv5 Huang et al., 2017) is available at https://doi.org/10.7289/V5T72FNM, and ERA5 Reanalysis (European Centre for Medium-Range Weather Forecasts, 2017, updated monthly) is available at https://doi.org/10.5065/D6X34W69. Post-processed CESM data used for the study is available at https://doi.org/10.5281/zenodo.

5513689. Complete CESM1.2.2 data are locally stored on the oschgerstore provided by the Oeschger Center for Climate Change Research (OCCR) and are available by request.

*Author contributions.* WMK, RB and CCR contributed to the design of the study. WMK carried out the climate simulations, performed the analysis and wrote the first draft. MS provided the volcanic forcing data. MM provided critical feedback on the results and drafted the manuscript together with WMK. All authors contributed to the writing and scientific discussion.

*Competing interests.* The authors declare no competing interests.

*Acknowledgements.* WMK and CCR are supported by the Swiss National Science Foundation (SNF) within the projects grant numbers: 200020_172745 and 200020_200492. MS received funding from the European Research Council under the European Union's Horizon 2020 research and innovation programme (grant agreement no. 820047). The simulations were performed on the supercomputing architecture of the Swiss National Supercomputing Centre (CSCS). Data is locally stored on the oschgerstore provided by the Oeschger Center for Climate
Change Research (OCCR). The ERSST v5 data are provided by the NOAA/OAR/ESRL PSL, Boulder, Colorado, USA, from their website at https://psl.noaa.gov/. The ERA5 data are provided by Copernicus Climate Change Service Climate Data Store (CDS), from their website at https://cds.climate.copernicus.eu. The authors acknowledge the help from Dr. Mathew Toohey for his responses and comments regarding the EVA model, Dr. Yafang Zhong for his comment on the implementation of the volcanic forcing in CESM, and Patricio Velasquez for his comments during the production of the volcanic forcing file.

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

**Table 1.** Forcing values for the simulations. The time-varying forcing in the last column is presented in Fig. 1.

| | | spin-up simulation | orbital-only simulation | full-forcing simulation |
|---|---|---|---|---|
| **time** | [years] | 1405 | 3351 (1501 BCE – 2008 CE) | 3351 (1501 BCE – 2008 CE) |
| **GHG** | | | | |
| $CO_2$ | [ppm] | 274.21 | 274.21 | Bereiter et al. (2015) and Meinshausen et al. (2017) |
| $CH_4$ | [ppb] | 572.88 | 572.88 | Joos and Spahni (2008) and Meinshausen et al. (2017) |
| $N_2O$ | [ppb] | 262.79 | 262.79 | Joos and Spahni (2008) and Meinshausen et al. (2017) |
| **VOL** | [Tg] | no forcing | no forcing | Carn et al. (2016) and Sigl et al. (2021) |
| **TSI** | [$Wm^{-2}$] | 1360.38 | 1360.38 | Vieira et al. (2011) and Usoskin et al. (2014) |
| **LULUC** | | 850 CE conditions | 850 CE conditions | Pongratz et al. (2008) |

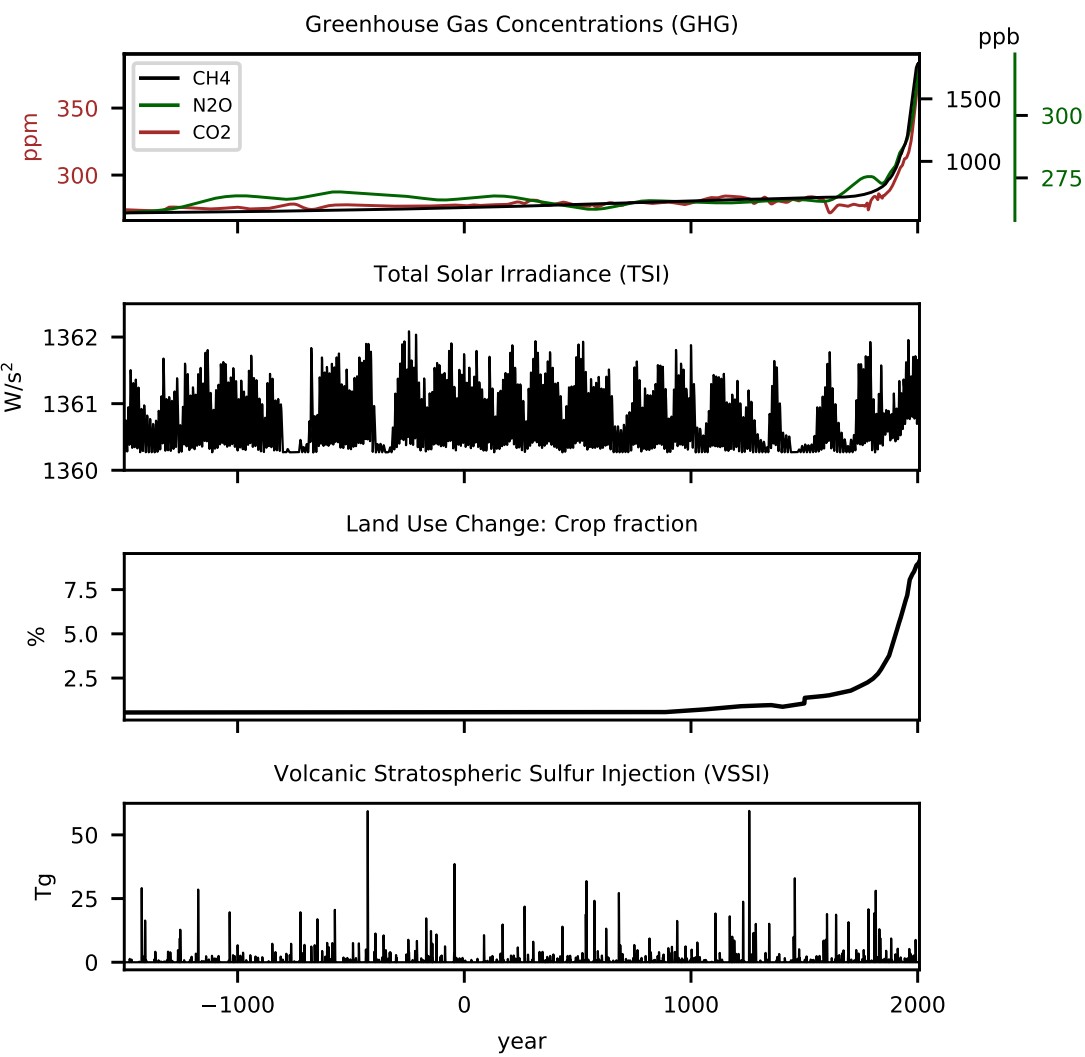

**Figure 1.** Time series of the external forcing included in the transient full-forcing simulation for 1501 BCE – 2008 CE.

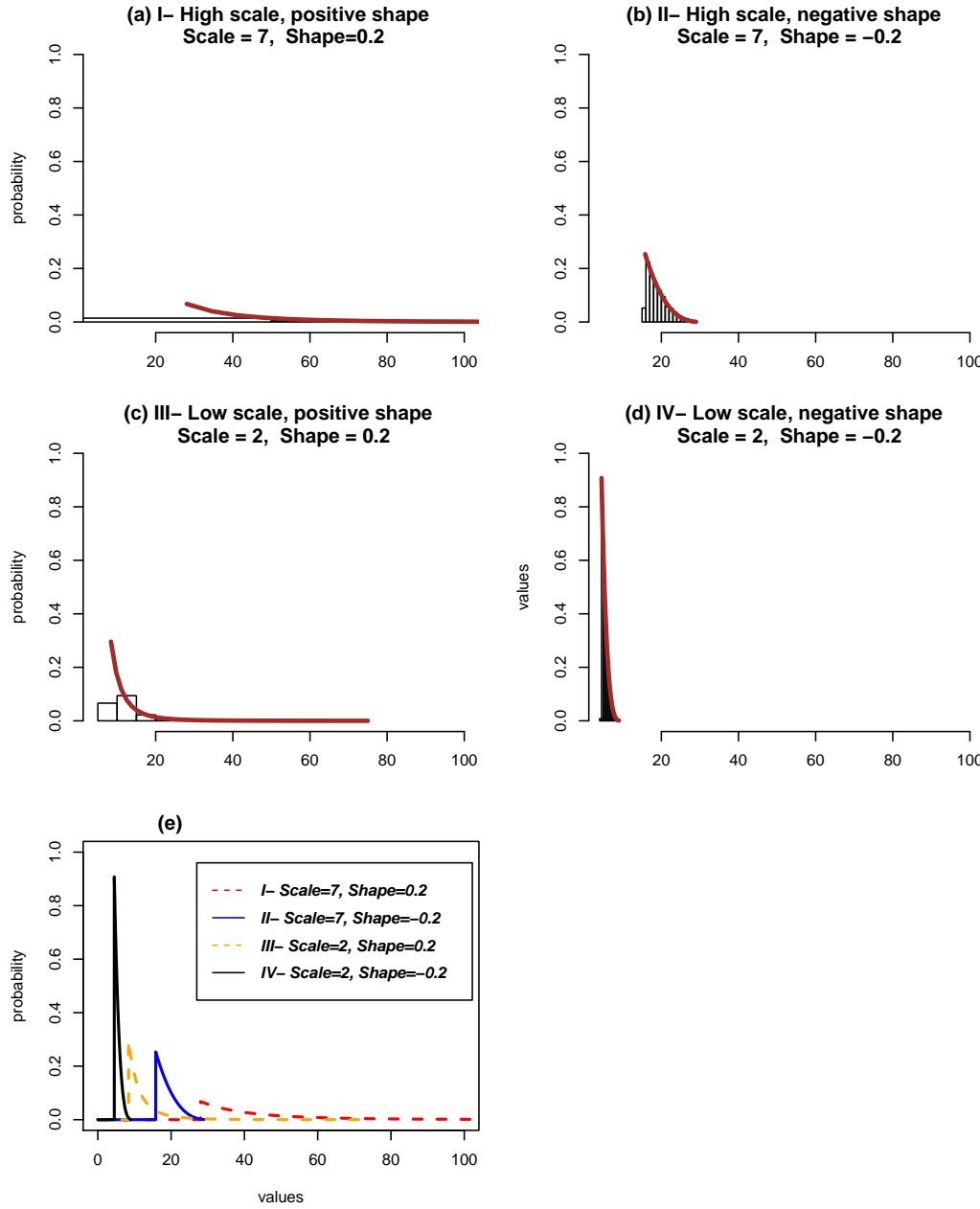

**Figure 2.** (a)-(d) Probability density distributions of extremes of a variable with different scale and shape parameters. (e) All distribution functions together.

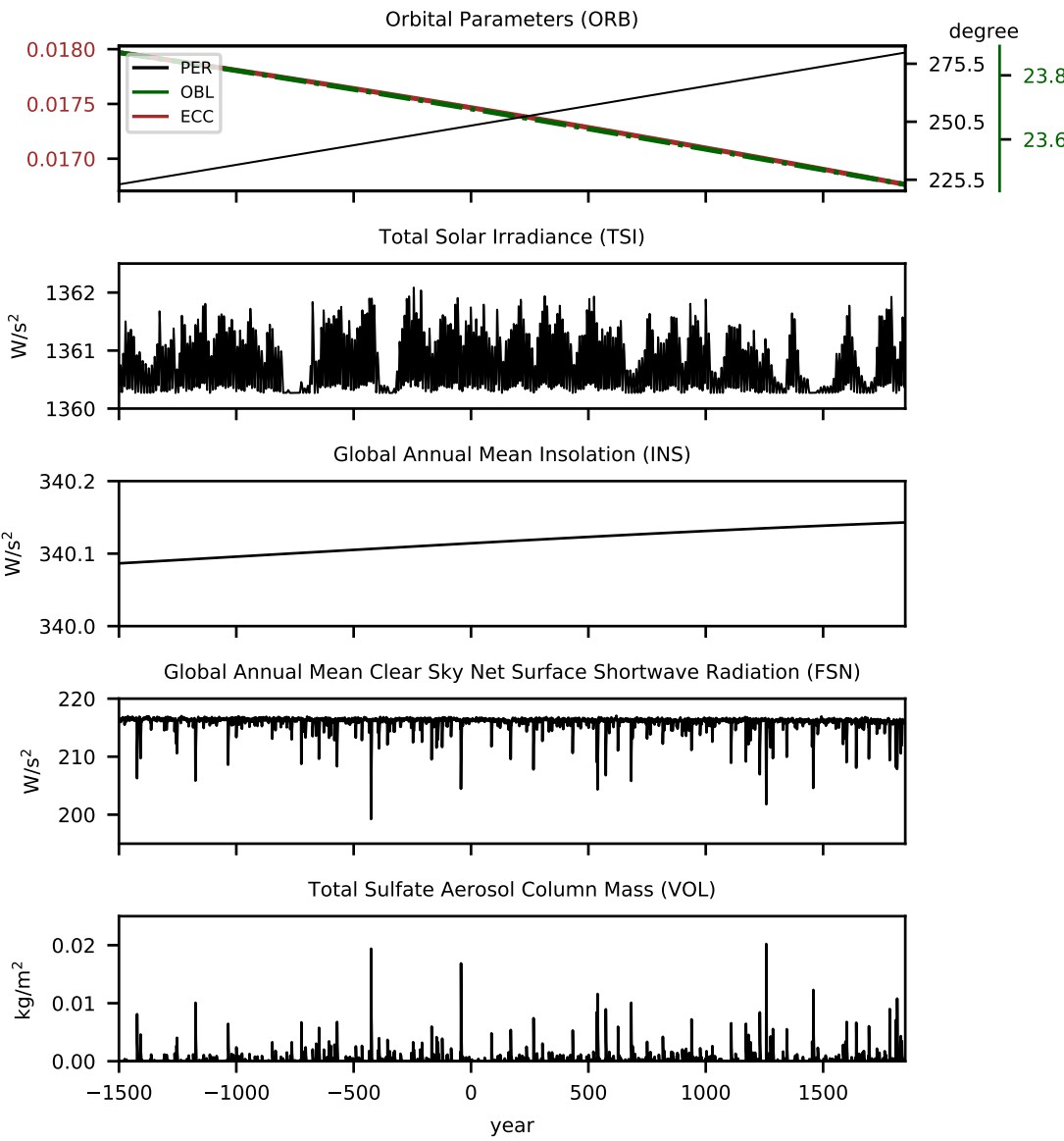

**Figure 3.** Time series of externally forced variability included as covariates in the non-stationary GPD models for the orbital-only and full-forcing simulations. Note that the VOL-covariate GPD model is only generated for the full-forcing simulation.

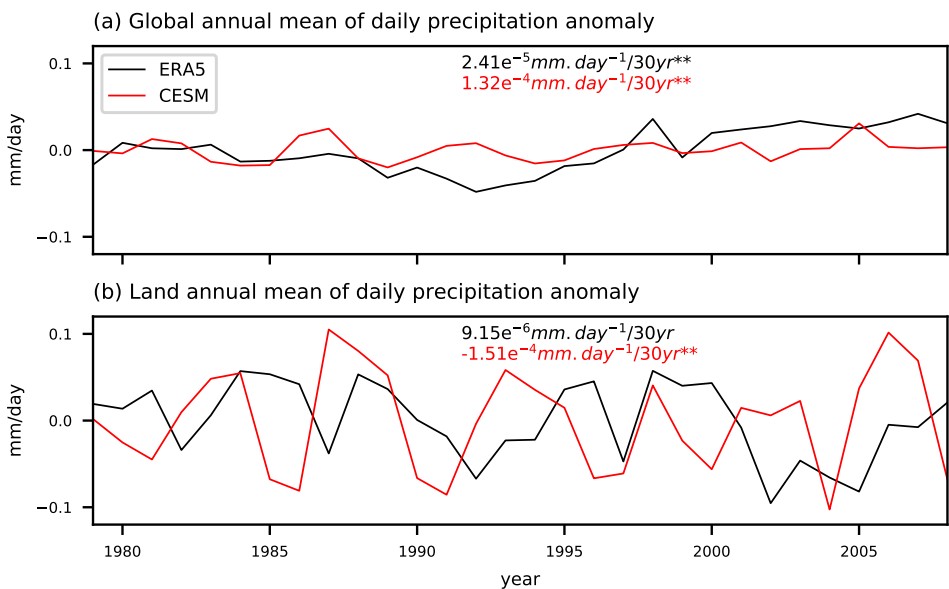

**Figure 4.** Time series of annually-averaged mean daily precipitation over the (a) entire globe and (b) land-only in ERA5 (black) and CESM (red). The values on the figures indicate the trends and ** denote that the trends are statistically different to zero at the 99% confidence interval based on the Mann–Kendall trend tests.

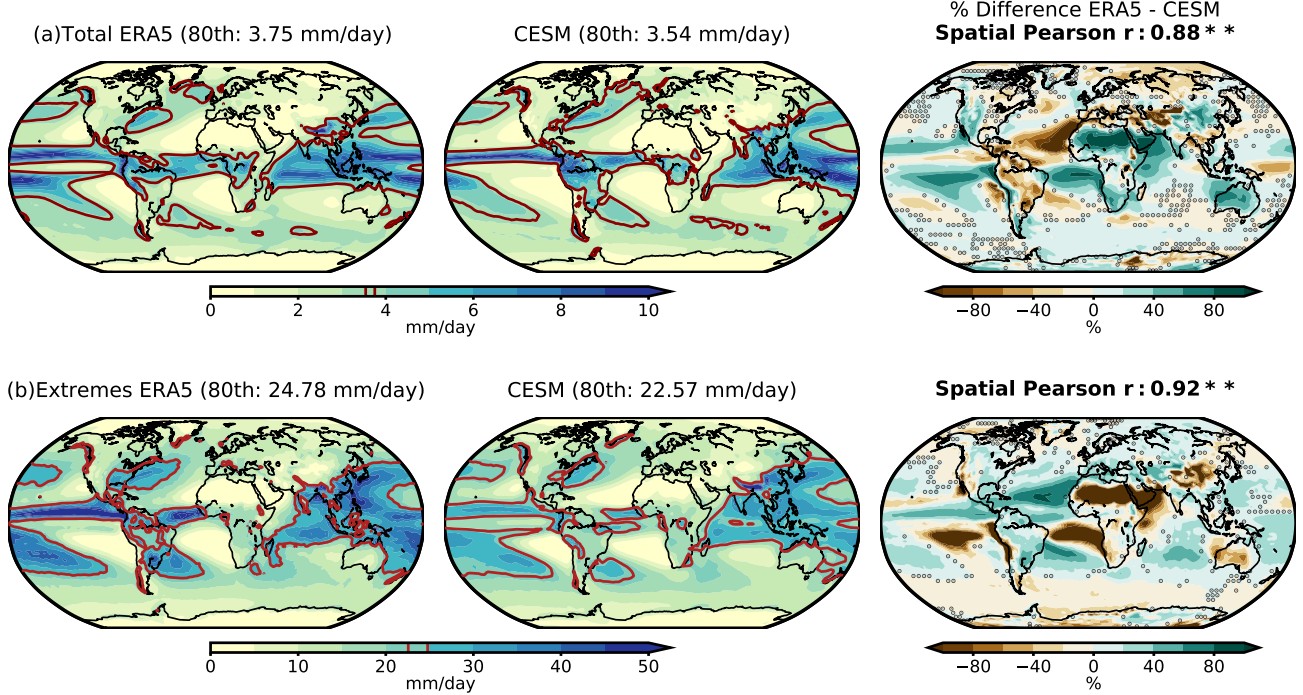

**Figure 5.** Mean values of ERA5 and CESM, and the percentages of difference between both datasets calculated as (ERA5 - CESM)/ERA5 for (a) the total daily precipitation and (b) the extreme daily precipitation (daily precipitation above the 95th percentiles relative to 1979–2008CE distributions). The brown line indicates the $80^{th}$ percentile level to discern wet regions (above the $80^{th}$ percentile) from others. Dotted regions in the difference plots indicate where the distributions of total or extreme precipitation are statistically similar at the 99% confidence interval based on the M-W U-tests. Pearson $r$ coefficients between the spatial mean values of ERA5 and CESM are denoted in bold on the difference plots accompanied with ** when the $r$ are statistically significant at the 99% confidence interval.

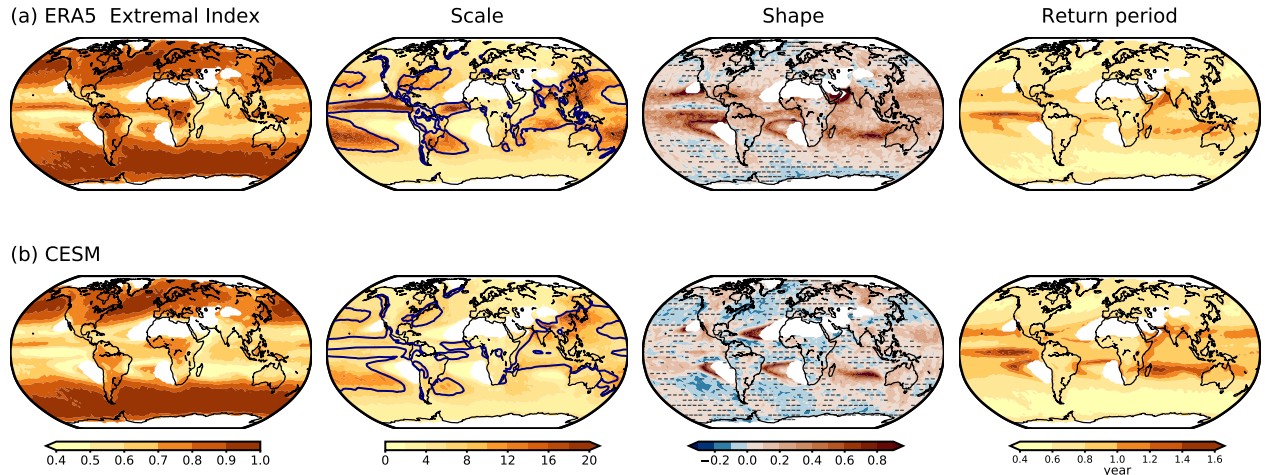

**Figure 6.** Extremal indices, parameters of the stationary GPD models, and return periods of the extreme thresholds in Fig. 5 for (a) ERA5 and (b) CESM. Striped regions indicate where the standard errors from the parameter estimation are higher than the estimated values. The dark blue line on the composite of the scale parameters denotes wet regions in Fig. 5. The regions where the annual total precipitation is less than 250 mm are excluded from the analysis and marked as white.

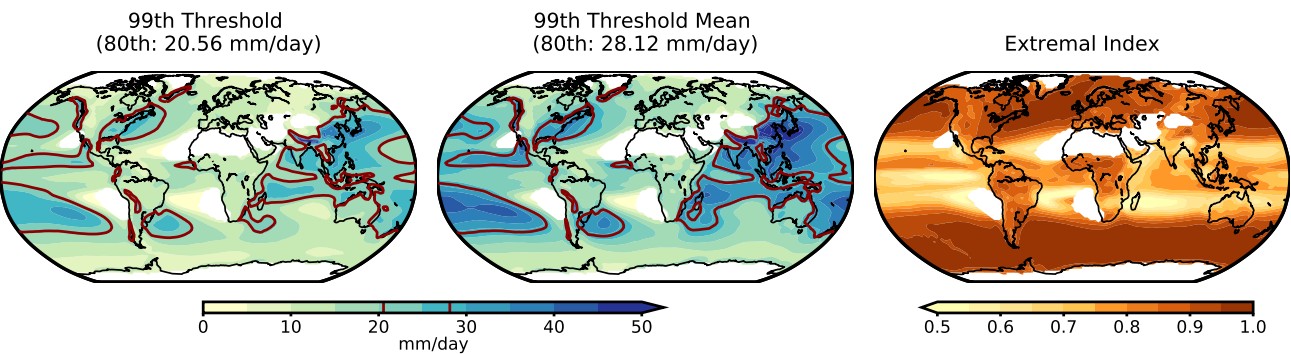

**Figure 7.** The thresholds for extreme precipitation (defined as the 99[th] percentiles of daily precipitation relative to the 3351-year distributions), means of the values above these thresholds, and extremal indices in the 1501 BCE–1849 CE full-forcing simulation. Brown line indicates the spatial 80[th] percentile level to discern wet regions (above the 80[th] percentile) from others.

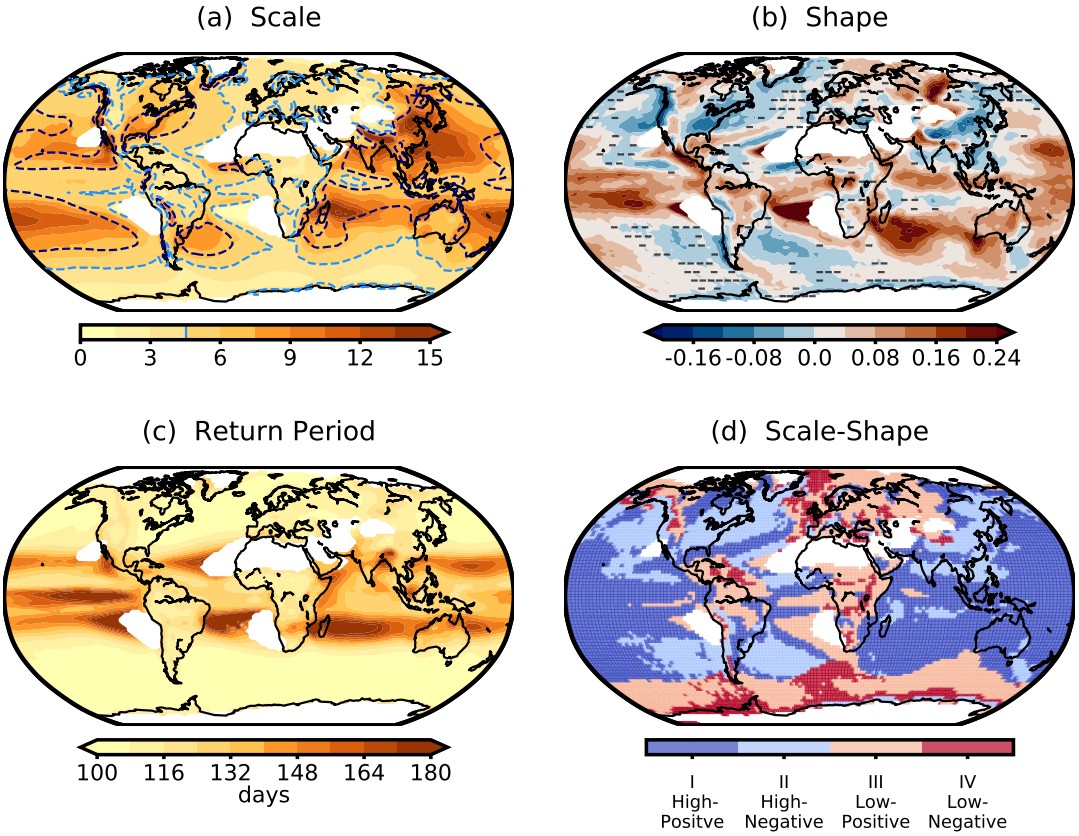

**Figure 8.** (a) Scale, (b) shape parameters of the stationary GPD models, (c) return periods of extreme thresholds (see Fig. 7), and (d) the combination of the scale and shape parameters with: above (High) and below (Low) the median of the scales, and positive and negative shape parameters. On the composite of scale parameters in (a), the wet region in Fig. 7 is denoted as dashed dark blue line, and the median of scale parameters is overlaid as dashed light blue line. Again, the regions where the annual total precipitation is less than 250 mm are marked as white.

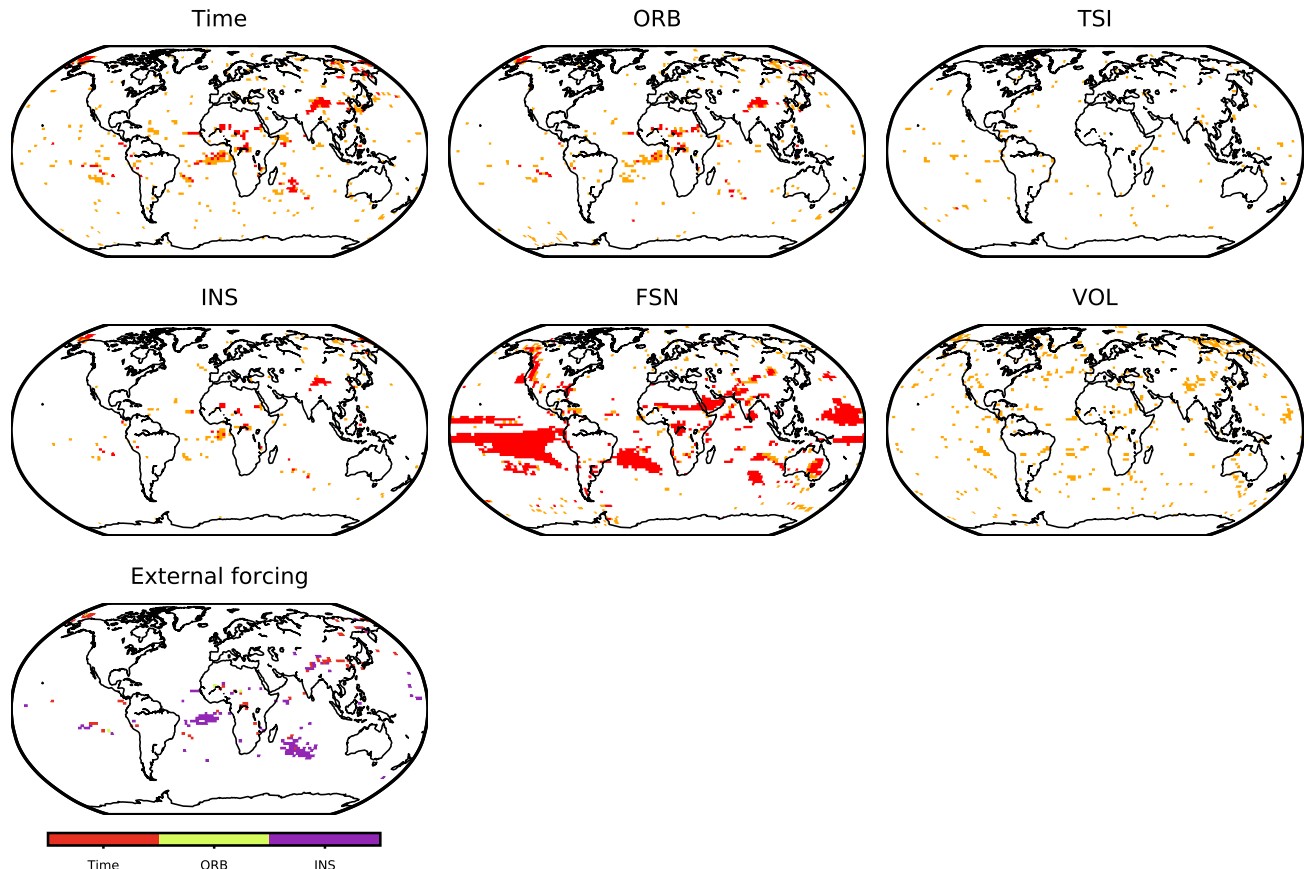

**Figure 9.** The external-forcing-GPD models that outperform the stationary GPD models at the 99% confidence interval based on the $D$ statistics and explain better the variability of extreme precipitation. Orange shades indicate the regions where the $D$ statistics in the full-forcing simulation are statistically significant and red shades where both full-forcing and orbital-only simulations present significant $D$ statistics. Note that the VOL-covariate model is not generated for the orbital-only simulation. The corresponding plots for each of the simulations can be found in the supplement Figs. S5 and S6. The lower panel of external forcing shows the best model among time-, ORB- and INS-covariate models by comparing the $D$ statistics among these three models.

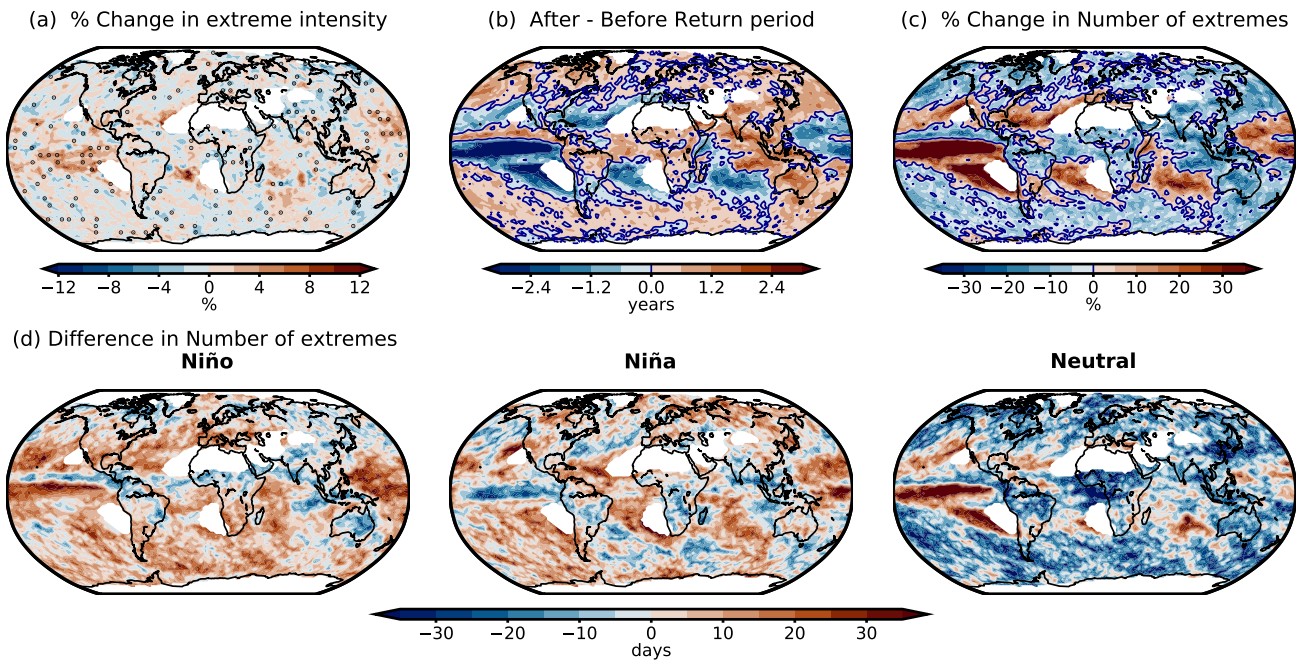

**Figure 10.** (a) Rates of change in the mean extreme precipitation, (b) differences in return periods, and (c) rates of change in the number of extreme precipitation during the period of three years after the tropical volcanic eruptions relative to three years before the tropical eruptions. The dotted regions in (a) indicate where the distributions of extreme precipitation between the pre- and post-eruption periods are statistically similar at the 99% confidence interval based on the M-W U-test. (d) Rates of change in the number of extremes in different ENSO states during the years of tropical eruptions (year 0) relative to the same ENSO states during the year before the tropical eruption.

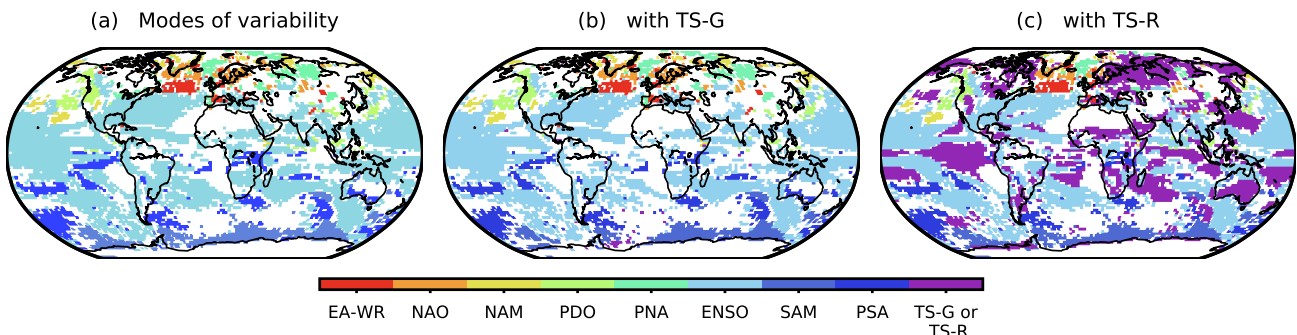

**Figure 11.** The non-stationary GPD models with (a) the modes of variability, (b) the modes of variability and TS-G, and (c) the modes of variability and TS-R as covariates that outperform all other GPD models (the modes-of-variability, TS-G or TS-R, and stationary GPD models) at 99% confidence interval, hence, explain best the variability of extreme precipitation. The regions where both orbital-only and full-forcing simulations share the same statistically significant $D$ statistics are shown. The plots that correspond to each of the simulations can be found in the supplement Fig. S7.