# Peer review of "Statistical characteristics of extreme daily precipitation during 1501 BCE – 1849 CE in the Community Earth System Model"

_Climate of the Past, 2021_

## Referee Comment (RC1)

**Review on "Statistical characteristics of extreme daily precipitation during 1501 BCE-1849 CE in the CESM**

June 17, 2021

The manuscript is clear, well-written, and provides a new approach to the understanding of extreme precipitation, and I recommend it for publication with minor revisions.

The authors introduce a methodology to analyze the characteristics of extreme precipitation in the past 3000 years, using peak-over-threshold extreme value analysis and the identification of modes of natural variability using EOF decompositions in distinct geographical areas. The adequacy of the model was first verified over the recent observational period against reanalysis data, before focusing on simulations of the pre-industrial era, with and without external forcing; and the statistical analysis then allows to link the characteristics of regional extreme events to internal variability and different types of forcing.

**1 Minor comments**

- L27: could cite Allen and Ingram (2002) for the different drivers of mean and extreme precipitation.

- L28 (rich gets richer): could cite Chou et al. (2009); Chou and Neelin (2004)

- L30 "is constrained by the available atmospheric moisture": to be more precise, it is constrained by the maximum available moisture content at a given temperature ($q_v^\star$ is what Clausius-Clapeyron knows about, not $q_v$), but global extremes occur in regions close to saturation, so it does make much difference for global extremes. It could make a difference for regional extremes, in regions where the values of rain percentiles are rather constrained by the relative occurence of different precipitation regimes.

- L72: you mention some model limitations later, but at this stage what is coming to mind is that you can add references for that fact that in CMIP5, the rich-gets-richer mechanism breaks in the torpics (Chadwick et al., 2013) and references for the drizzling bias (e.g. Stephens and Hu, 2010)

- L84: interesting

- L110: would there be any interest in having several ensemble members for the robustness of attribution of extremes to different modes of variability?

- L120: Is there any uncertainty reported on the forcing data, that could propagate to uncertainties in extreme rainfall characteristics?

- L147, section 3.1: extremely clear, well-explained

- L168 "each cluster is composed of consecutive days of extremes": which threshold is used to separate the clusters, how is it chosen, and is it spatially uniform?

- L181: I am bit confused, it is not clear to me how the data is sampled for the calculation of a given non-stationary GPD. For stationary GPDs, I thought you were using the entire time series at each location and computed the shape, scale parameters for that whole distribution over time. For non-stationary distributions, it seems that you somehow fit $\sigma_0$ and $\sigma_1$ for the entire dataset, but how do you get the evolution of the tail distribution over time for each $(x, y)$ point? Do you compute the rain distribution and the GPD for data sampled on moving windows, then obtain a $\sigma(t)$ and regress it over time to get $\sigma_0$ and $\sigma_1$?

- L244 (Mann-Kendall trend test): I don't know this test, would you have a reference? Is it standard practice?

- L246 (Mann-Whitney U-test): I don't know this test, would you have a reference? Is it standard practice?

- L269: very interesting

- L284 "the de-clustering method *is* (typo) only around 40% of the initial numbers of extremes": same comment as above, does that depend on the threshold used to define clusters? Is it uniform?

- L286: interesting comparison of the extremal index between tropical and extra-tropical regions. It seems there could be a correspondence with the convective-organization viewpoint, saying that organization is more likely to occur at high SSTs, and which would be consistent with clusters mainly occurring in the tropics.

- L350-351 "distinguishing *the* regions": do you mean that 'the POT analysis allows to separate distinct and coherent contiguous regions for similar types of distributions', or do you mean that the regions that are exhibited somehow map onto known regimes, ie the regions on Fig. 5b?

- Figure 2: What would the distributions look like with a logarithmic y-axis? Maybe that would allow to better illustrate when there is a finite upper-bound. Is it hard to see as it is shown now.

- Figure 7, middle panel title: "mean above 99th threshold" would me more explicit.

- Appendix A, L761-762: several variables are used and I assume they are normalized with mean and variance before analysis so that the RMSE can be compared numerically. But is there a reference RMSE value, a threshold above/below which the error is large/small, or is it just for relative comparison across modes?

**References**

M. R. Allen and W. J. Ingram. Constraints on Future Changes in Climate and the Hydrologic Cycle. *Nature*, 419(6903):224–232, sep 2002. ISSN 0028-0836. doi: 10.1038/nature01092. URL http://www.ncbi.nlm.nih.gov/pubmed/12226677.

Robin Chadwick, Ian Boutle, and Gill Martin. Spatial patterns of precipitation change in CMIP5: Why the rich do not get richer in the tropics. *Journal of Climate*, 26(11):3803–3822, 2013. ISSN 08948755. doi: 10.1175/JCLI-D-12-00543.1.

Chia Chou and J David Neelin. Mechanisms of Global Warming Impacts on Regional Tropical Precipitation. *Journal of Climate*, 17:2688–2701, 2004.

Chia Chou, J David Neelin, Chao-An Chen, and Jien-Yi Tu. Evaluating the "Rich-Get-Richer" Mechanism in Tropical Precipitation Change under Global Warming. *Journal of Climate*, 22(8):1982–2005, apr 2009. ISSN 0894-8755. doi: 10.1175/2008JCLI2471.1. URL http://journals.ametsoc.org/doi/abs/10.1175/2008JCLI2471.1.

Graeme L Stephens and Yongxiang Hu. Are Climate-Related Changes to the Character of Global-Mean Precipitation Predictable? *Environmental Research Letters*, 5(2):025209, apr 2010. ISSN 1748-9326. doi: 10.1088/1748-9326/5/2/025209. URL `http://stacks.iop.org/1748-9326/5/i=2/a=025209?key=crossref.d5277cd96f73aefba73aad997e4af21a`.

---

## Author Comment (AC1)

Manuscript cp-2021-61

**Response to the reviewer 1**

We would like to thank the reviewers for their constructive feedbacks and insightful comments. We greatly appreciate the time and effort you dedicated to revise our manuscript, which helped us to improve our presentation. We have incorporated your suggestions in the revised manuscript, and you can find our responses (in blue) below.

**Comments:**

*1) L27: could cite Allen and Ingram (2002) for the different drivers of mean and extreme precipitation.*

*2) L28 (rich gets richer): could cite Chou et al. (2009); Chou and Neelin (2004)*

*3) L30 "is constrained by the available atmospheric moisture": to be more precise, it is constrained by the maximum available moisture content at a given temperature ($q_v^\star$ is what Clausius-Clapeyron knows about, not $q_v$), but global extremes occur in regions close to saturation, so it does make much difference for global extremes. It could make a difference for regional extremes, in regions where the values of rain percentiles are rather constrained by the relative occurence of different precipitation regimes.*

*4) L72: you mention some model limitations later, but at this stage what is coming to mind is that you can add references for that fact that in CMIP5, the rich-gets-richer mechanism breaks in the torpics (Chadwick et al., 2013) and references for the drizzling bias (e.g. Stephens and Hu, 2010)*

**Responses to the comments 1 to 4: We will include the suggested citations and correct the line 30 as suggested.**

*5) L110: would there be any interest in having several ensemble members for the robustness of attribution of extremes to different modes of variability?*

**Thanks for the idea. Using a large ensemble would be always ideal to assess the robustness of modes of internal variability in climate simulations. However, there is a technical constraint on performing several ensemble members of these 3351 year-long simulations with daily time resolution. In case of CMIP5, very few (only two) simulations are available at daily time resolutions covering the past millennium.**

**Nevertheless, as our analysis considered all the possible conditions of the modes of variability linked with a large number of daily extreme precipitation events during the entire 3351 years, we assume that this amount of data can partially increase the robustness of the association of extreme precipitation with the modes of variability in CESM.**

*6) L120: Is there any uncertainty reported on the forcing data, that could propagate to uncertainties in extreme rainfall characteristics?*

**Forcing data have for sure some inherent biases coming from the reconstructions. However, based on the studies on the present-day and future extreme precipitation, extreme precipitation is more dependent on the surface temperature (Pendegrass et al., 2015; Sillmann et al., 2017) and internal variability (Sillmann et al., 2017), rather than the introduced external forcing, for instance, the emission scenarios and solar forcing. Hence, we assume that the uncertainty in extreme rainfall characteristics caused by the forcing data would be minimal.**

*7) L168 "each cluster is composed of consecutive days of extremes": which threshold is used to separate the clusters, how is it chosen, and is it spatially uniform?*

**The temporal threshold is set to the distance between the extremes within a cluster. In our study, the value of this threshold is one day. In other words, the maximum temporal distance allowed among the extremes in one cluster is one day, hence, the minimum distance among the clusters is two days. This temporal threshold of one day is a commonly used value in many GEV analysis (Coles et al, 2001; Sugahara et al., 2008), as it does not significantly reduce the number of data to be analyzed and at the same time, it guarantees a statistical independence among the de-clustered extremes. We do not necessarily expect that the clusters are uniformly distributed over the space and time, as except the temporal threshold of one day within a cluster (thus, two days among the clusters), there is no other spatial restriction imposed on the clusters.**

**We will reformulate the corresponding sentence in the revised manuscript to make it clearer.**

*8) L181: I am bit confused, it is not clear to me how the data is sampled for the calculation of a given non-stationary GPD. For stationary GPDs, I thought you were using the entire time series at each location and computed the shape, scale parameters for that whole distribution over time. For non-stationary distributions, it seems that you somehow fit σ0 and σ1 for the entire dataset, but how do you get the evolution of the tail distribution over time for each (x,y) point? Do you compute the rain distribution and the GPD for data sampled on moving windows, then obtain a σ(t) and regress it over time to get σ0 and σ1?*

**The GEV analysis does not consider the entire distributions, but it only works with the tails of the distributions. Thus, only the extremes, which are located at the tail of the distribution, are fit to σ(t) = σ0 + σ1*C(t). The fit to get the scale parameters σ(t) at each (x,y) and at time *t* is done through the maximum likelihood estimation.**

**The basis of maximum likelihood estimation for a non-stationary GPD is as follows (Coles et al., 2001):**

**The complete vector of GPD parameters β is β=[σ(t), ξ] as only the scale parameter σ varies with time. Under the assumption that the extremes $z_1$, …, $z_m$ at *t*=1,..,m are independent variables, the log-likelihood function for the parameters σ and ξ when ξ ≠ 0 is:**

$$l(\sigma(t), \xi) = - \sum_{t=1}^{k} \left\{ \log(\sigma(t)) + \left(1 + \frac{1}{\xi}\right) log\left[1 + \frac{\xi z_i}{\sigma(t)}\right] \right\}$$

**And when ξ = 0, the approximation that ξ → 0 is used and the log-likelihood function becomes:**

$$l(\sigma(t), \xi) = - \sum_{t=1}^{k} \left\{ \log(\sigma(t)) + z_i \sigma(t)^{-1} \right\}$$

**Where σ(t) is σ0 + σ1*C(t) and k is the number of (de-clustered) extremes above threshold.**

**Then, the maximization of the pair of log-likelihood l(σ(t), ξ) with respect to the parameter vector β=[σ(t), ξ] is performed and this leads to the maximum likelihood estimate of the parameters σ(t) and ξ. For σ(t), β is [σ0, σ1, ξ] (Adlouni et al., 2007). This maximization is done numerically, as no analytical solution is possible.**

**In the revised manuscript, we will include this explanation on the estimation of the parameters (Note that the parameters for the stationary GPD are also estimated with the same method, but considering the constant σ.)**

*9) L244 (Mann-Kendall trend test): I don't know this test, would you have a reference? Is it standard practice?*

*10) L246 (Mann-Whitney U-test): I don't know this test, would you have a reference? Is it standard practice?*

**Response to the comments 10 and 11: In the revised manuscript, we will add some references to support the tests we used.**

*11) L284 "the de-clustering method is (typo) only around 40% of the initial numbers of extremes": same comment as above, does that depend on the threshold used to define clusters? Is it uniform?*

**Also refer to our response #8. The final number of de-clustered extremes depends on the sizes of clusters which are defined by the temporal threshold. If the temporal threshold among the extremes is softened (for example, instead of 1-day distance, taking 5-day distance), the size of each cluster would be large, and the number of de-clustered extremes would be reduced as the de-clustering method only takes a maximum extreme in each cluster.**

*12) L286: interesting comparison of the extremal index between tropical and extra-tropical regions. It seems there could be a correspondence with the convective-organization view-point, saying that organization is more likely to occur at high SSTs, and which would be consistent with clusters mainly occurring in the tropics.*

**Thanks very much for your comment. We agree with you that more consistent clustered convective organization would likely occur more at high SST and over the ocean where sufficient sources of moisture are available.**

*13) L350-351 "distinguishing the regions": do you mean that 'the POT analysis allows to separate distinct and coherent contiguous regions for similar types of distributions', or do you mean that the regions that are exhibited somehow map onto known regimes, ie the regions on Fig. 5b?*

**It means the former. We will reformulate the corresponding sentence clarifying the point in the revised manuscript.**

*14) Figure 2: What would the distributions look like with a logarithmic y-axis? Maybe that would allow to better illustrate when there is a finite upper-bound. Is it hard to see as it is shown now.*

**We will modify the figure 2 in the revised manuscript.**

*15) Figure 7, middle panel title: "mean above 99th threshold" would me more explicit.*

**We will modify the part of the caption as: "The thresholds for extreme precipitation defined as the 99th percentiles of daily precipitation relative to the 3351-year distributions."**

*16) Appendix A, L761-762: several variables are used and I assume they are normalized with mean and variance before analysis so that the RMSE can be compared numerically. But is there a reference RMSE value, a threshold above/below which the error is large/small, or is it just for relative comparison across modes?*

**Yes, the variables are normalized and standardized before the calculation. As you said, it is the relative comparison across the modes. Although, we did not mention in the manuscript, we excluded a EOF pattern from the model that theoretically should agree with the observed EOF, but present high RMSE relative to the observed pattern.**

**In addition, we selected the modes that are detected as well in the study by Fasullo et al. (2020) which used the CMIP models including the CESM family models, and by Lim (2014) which used the reanalysis. Both studies estimated the modes of variability based on the EOF analysis.**

**In the revised manuscript, we will mention these two literatures we have based on.**

**References**

El Adlouni, S., Ouarda, T.B., Zhang, X., Roy, R. and Bobée, B., 2007. Generalized maximum likelihood estimators for the nonstationary generalized extreme value model. Water Resources Research, 43(3), https://doi.org/10.1029/2005WR004545

Coles, S., Bawa, J., Trenner, L. and Dorazio, P., 2001. An introduction to statistical modeling of extreme values (Vol. 208, p. 208). London: Springer.

Fasullo, J.T., Phillips, A.S. and Deser, C., 2020. Evaluation of leading modes of climate variability in the CMIP archives. Journal of Climate, 33(13), pp.5527-5545, https://doi.org/10.1175/JCLI-D-19-1024.1

Lim, Y.K., 2015. The East Atlantic/West Russia (EA/WR) teleconnection in the North Atlantic: climate impact and relation to Rossby wave propagation. Climate Dynamics, 44(11-12), pp.3211-3222, link= https://link.springer.com/content/pdf/10.1007/s00382-014-2381-4.pdf

Pendergrass, A.G., Lehner, F., Sanderson, B.M. and Xu, Y., 2015. Does extreme precipitation intensity depend on the emissions scenario?. Geophysical Research Letters, 42(20), pp.8767-8774, https://doi.org/10.1002/2015GL065854

Sillmann, J., Stjern, C.W., Myhre, G. and Forster, P.M., 2017. Slow and fast responses of mean and extreme precipitation to different forcing in CMIP5 simulations. Geophysical Research Letters, 44(12), pp.6383-6390, https://doi.org/10.1002/2017GL073229

Sugahara, S., Da Rocha, R.P. and Silveira, R., 2009. Non-stationary frequency analysis of extreme daily rainfall in Sao Paulo, Brazil. International Journal of Climatology: A Journal of the Royal Meteorological Society, 29(9), pp.1339-1349, https://doi.org/10.1002/joc.1760

---

## Author Response (AR1)

**Manuscript cp-2021-61**

Dr. Ran Feng, editor Climate of the Past

Woon Mi Kim Physics Institute and Oeschger Centre for Climate Change Research, University of Bern Sidlerstrasse 5, 3012 Bern, Switzerland August 11, 2021

Dear Dr. Ran Feng,

We would like to thank you again for the opportunity to resubmit our manuscript and also many thanks to the reviewers for their constructive feedbacks and insightful comments. We greatly appreciate the time that you and the reviewers dedicated to our manuscript. We have addressed all the reviewer's comments carefully. Please find below our responses (in bold blue fonts) to the reviewers.

In addition, note that we included some sentences about the forcing data we used, which were omitted during the first phase of the revision. We marked-up these inclusions in the revised manuscript (lines 125 - 126 and lines 128 - 129).

Sincerely,

Woon Mi Kim on behalf of all the authors

**Response to reviewer 1**

**Comments:**

1) L27: could cite Allen and Ingram (2002) for the different drivers of mean and extreme precipitation.

We included the corresponding citation in lines 31, 36, 41 and 45.

2) L28 (rich gets richer): could cite Chou et al. (2009); Chou and Neelin (2004)

We included the corresponding citations in line 33.

3) L30 "is constrained by the available atmospheric moisture": to be more precise, it is constrained by the maximum available moisture content at a given temperature ( $qv \star$  is what Clausius-Clapeyron knows about, not qv), but global extremes occur in regions close to saturation, so it does make much difference for global extremes. It could make a difference for regional extremes, in regions where the values of rain percentiles are rather constrained by the relative occurrence of different precipitation regimes.

In the revised manuscript, we included some sentences clarifying the corresponding part in lines 39 - 41 and 43 - 45.

Lines 39 – 45: "[...] changes in extreme precipitation are constrained by the available maximum low-level atmospheric moisture at a given temperature following the Clausius–Clapeyron (C-C) relationship (Allen and Ingram, 2002; Pall et al., 2007). The reasoning is as follows: the low-level atmosphere can hold more moisture with increasing temperatures, which in turn leads to an increase in extreme precipitation (Trenberth et al., 2003; Pall et al., 2007; Fischer and Knutti, 2016). The rate of increase of extreme precipitation given by C-C is 6-7% per degree of warming. This relationship holds mostly true over higher latitudes where the air is usually closer to saturation and relative humidity is roughly constant (Allen and Ingram, 2002)."

4) L72: you mention some model limitations later, but at this stage what is coming to mind is that you can add references for that fact that in CMIP5, the rich-gets-richer mechanism breaks in the tropics (Chadwick et al., 2013) and references for the drizzling bias (e.g. Stephens and Hu, 2010)

Thank you very much for your suggestion. We included more about the limitations related to the "rich-gets-richer" mechanism in the initial part of the introduction in lines 34 – 38, also including Chadwick et al. (2013). However, we did not include Stephens and Hu (2010) as the drizzling bias is more related to the changes in global mean precipitation and its associated water vapor and water-recycling, and it is not specifically related to the model limitations to represent extreme precipitation.

Lines 34 – 38: "The "wet gets wetter, and dry gets drier" pattern denotes the intensification of the global hydrological cycle, which is controlled by a tropospheric energy budget (Boer, 1993; Allen and Ingram, 2002; Yang et al., 2003). Nevertheless, it is also noted that this pattern for the future mean precipitation is more heterogeneous over land areas in observations and climate models (Roderick et al., 2014; Byrne and O'Gorman, 2015) and breaks in the tropics in CMIP5 models (Chadwick et al., 2013)."

5) L110: would there be any interest in having several ensemble members for the robustness of attribution of extremes to different modes of variability?

Note that our response to this comment is the same as what we responded before during the discussion phase:

Thank you for this idea. Using a large ensemble would always be ideal for assessing the robustness of modes of internal variability in climate simulations. However, there is a technical constraint on performing several ensemble members of these 3351 year-long simulations with daily time resolution. In the case of CMIP5, very few simulations are available at daily time resolutions covering the past millennium.

Nevertheless, as our analysis considered all the possible conditions of the modes of variability linked with a large number of daily extreme precipitation events during the entire 3351 years, we assume that this amount of data can increase the robustness of the association of extreme precipitation with the modes of variability in CESM.

*6)* L120: Is there any uncertainty reported on the forcing data that could propagate to uncertainties in extreme rainfall characteristics?

The forcing data have for sure some inherent uncertainties coming from the reconstructions. However, based on the studies on the present-day and future extreme precipitation, extreme precipitation is more dependent on the surface temperature (Pendegrass et al., 2015; Sillmann et al., 2017) and internal variability (Sillmann et al., 2017), rather than the introduced external forcing, for instance, the emission scenarios and solar forcing. Hence, we assume that the uncertainty in extreme rainfall characteristics caused by the forcing data would be minimal.

We included some texts about uncertainties from the external forcings in lines 243 - 247 in the method section (Sect. 3.2) and in lines 491 - 493 in the result section (Sect. 4.4).

Lines 243 – 247: "It is important to mention that all reconstructed external forcings such as TSI and volcanic eruptions contain inherent uncertainties derived from reconstruction models or methods and from the dating of past events (Sigl et al., 2015; Jungclaus et al., 2017; Matthes et al., 2017). An attempt to reduce such uncertainties is an active research topic (Sigl et al., 2015; Matthes et al., 2017) that is beyond the scope of this study. A possible implication of uncertainties from the external forcings in our analysis is briefly discussed in the result section (Sect. 4.4)."

Lines 491 – 493: "Moreover, this limited influence of external forcings on extreme precipitation signifies that the inherent uncertainties of external forcings have a minimal effect on the characterization of pre-industrial extreme precipitation."

7) L168 "each cluster is composed of consecutive days of extremes": which threshold is used to separate the clusters, how is it chosen, and is it spatially uniform?

The temporal threshold is set to the distance between the extremes within a cluster. In our study, the value of this threshold is one day. In other words, the maximum temporal distance allowed among the extremes in one cluster is one day, hence, the minimum distance among the clusters is two days. This temporal threshold of one day is a commonly used value in many GEV analyses (Coles et al, 2001; Sugahara et al., 2008), as it does not significantly reduce the number of data to be analyzed. At the same time it guarantees statistical independence among the de-clustered extremes. We do not necessarily expect that the clusters are uniformly distributed over space and time.

We reformulated the corresponding sentence in the revised manuscript to make it clearer.

Lines 180 – 183: "For this, we de-cluster the extreme precipitation at each grid point by taking the maximum value within each cluster. Each cluster is composed of consecutive days of extremes and the extremes separated by a maximum of one day to other extremes. In other words, the minimum temporal distance allowed between the extremes within a cluster is one day, and between the clusters is two days (Coles et al., 2001)."

8) L181: I am bit confused, it is not clear to me how the data is sampled for the calculation of a given non-stationary GPD. For stationary GPDs, I thought you were using the entire time series at each location and computed the shape, scale parameters for that whole distribution over time. For nonstationary distributions, it seems that you somehow fit  $\sigma$ 0 and  $\sigma$ 1 for the entire dataset, but how do you get the evolution of the tail distribution over time for each (x,y) point? Do you compute the rain distribution and the GPD for data sampled on moving windows, then obtain a  $\sigma$ (t) and regress it over time to get  $\sigma$ 0 and  $\sigma$ 1?

As we mentioned in the previous discussion phase, we included the detail on the estimation method for the parameters  $\sigma$ ,  $\sigma 0$ ,  $\sigma 1$  and  $\xi$  through maximum likelihood in lines 189 – 196.

Lines 189 – 196: "The parameter estimation for a stationary GPD model through a maximum likelihood is given as follows: under the assumption that the exceedances  $z_i$ , ...,  $z_k$  are independent variables where k is the number of exceedances, the log-likelihood function *I* for the parameters  $\sigma$  and  $\xi$  is:

 $l(\sigma,\xi) = -k \log(\sigma) - \left(1 + \frac{1}{\xi}\right) \sum_{i=1}^{k} \log(1 + \frac{\xi z_i}{\sigma}) \quad \text{for} \quad (1 + \sigma^{-1}\xi z_i) > 0, \quad i=1,...,k$  $l(\sigma) = -k \log(\sigma) - \sigma^{-1} \sum_{i=1}^{k} z_i \quad \text{for} \quad \xi = 0$

Having the parameter vector  $\beta$  with  $\beta = [\sigma, \xi]$ , the maximization of the pair of log-likelihood  $l(\sigma, \xi)$  with respect to the  $\beta$  is performed. This maximization leads to the maximum likelihood estimate of the scale  $\sigma$  and shape  $\xi$ . The maximization is done numerically, as no analytical solution is possible (Coles et al., 2001)."

Lines 203 – 205: "Time dependent scale and shape parameters for the non-stationary model are also estimated using the maximum likelihood method following Eq. 3, assuming  $\sigma(t) = \sigma_0 + \sigma_1 C(t)$  or  $\sigma(t) = \sigma_0 + \sigma_1 t$  (Colesetal.,2001; Sugahara et al.,2009). The parameter vector  $\beta$  in this case is  $\beta = [\sigma_0, \sigma_1, \xi]$  (El Adlouni et al., 2007)."

*9)* L244 (Mann-Kendall trend test): I don't know this test, would you have a reference? Is it standard practice?

We added some references (Mann, 1945; Wilks, 2011; e.g., Wesler et al., 2013) in lines 280.

10) L246 (Mann-Whitney U-test): I don't know this test, would you have a reference? Is it standard practice?

We added some references (Mann and Whitney, 1947; Wilks, 2011; e.g., Kim and Raible, 2021) in lines 282. Note that nonparametric test such as the Mann-Whitney U-test are commonly used in particular for extreme events which are by definition not normal distributed (which is the assumption of the commonly used student t-test).

11) L284 "the de-clustering method is (typo) only around 40% of the initial numbers of extremes": same comment as above, does that depend on the threshold used to define clusters? Is it uniform?

The response for this comment is the same as what we responded during the discussion phase (see #7):

The number of de-clustered extremes depends on the size of the clusters that are defined by the level of temporal threshold. If the temporal threshold among the extremes is softened (for example, instead of 1-day distance, taking 5-day distance), the size of each cluster would be large, and the final de-clustered values would be reduced as the de-clustering method only takes a maximum extreme in each cluster.

12) L286: interesting comparison of the extremal index between tropical and extra-tropical regions. It seems there could be a correspondence with the convective-organization view- point, saying that organization is more likely to occur at high SSTs, and which would be consistent with clusters mainly occurring in the tropics.

Thank you for this hint. We added a sentence:

Lines 320 – 324: "In some regions in the tropics, the de-clustering method leaves only around 40% of the initial numbers of extremes, indicating that the reduction of extreme events is particularly strong over this latitudinal belt, a known region of convective organization with temporal clustering."

13) L350-351 "distinguishing the regions": do you mean that 'the POT analysis allows to separate distinct and coherent contiguous regions for similar types of distributions', or do you mean that the regions that are exhibited somehow map onto known regimes, i.e. the regions on Fig. 5b?

It means the former. We changed the corresponding sentence to:

Lines 388 – 389: "the results show that the POT analysis is able to identify the large-scale characteristics of extreme precipitation by separating distinct and coherent regions for similar types of distributions."

14) Figure 2: What would the distributions look like with a logarithmic y-axis? Maybe that would allow to better illustrate when there is a finite upper-bound. Is it hard to see as it is shown now.

We modified the figure 2 by including a plot that shows all the density distributions together (Figure 2.e). We did not change the y-axis to a logarithmic one, as the resulting plot with a logarithmic axis seemed less convincing than showing all the distribution together with the same x and y coordinate values.

15) Figure 7, middle panel title: "mean above 99th threshold" would me more explicit.

The part of the caption is modified as: "The thresholds for extreme precipitation (defined as the 99th percentiles of daily precipitation relative to the 3351-year distributions)"

16) Appendix A, L761-762: several variables are used and I assume they are normalized with mean and variance before analysis so that the RMSE can be compared numerically. But is there a reference RMSE value, a threshold above/below which the error is large/small, or is it just for relative comparison across modes?

Yes, the variables are normalized and standardized before the calculation. As you said, it is the relative comparison across the modes. Although we did not mention it in the manuscript, we excluded an EOF pattern from the model that theoretically should agree with the observed EOF, but it presents high RMSE relative to the observed pattern.

Besides the criterion based on RMSE, we selected the modes that are similar and detected as well by Fasullo et al. (2020) which used the CMIP models including the CESM family models, and by Lim (2014) which used the reanalysis. Both studies estimated the modes of variability based on the EOF analysis. In the revised manuscript, we mention the two studies we have based the analysis on.

Also, note that in the revised version, we moved all information we put before in the appendix to the supplement.

**Response to reviewer 2**

**Major comment:**

1) Main suggestions is related to the link between temperature and extreme precipitations. Whereas the manuscript focused on the external drivers of extreme precipitation and on the impact of natural modes of climate variability, one question that probably will come to the mind of many readers is whether or not warm decadal (or miltidecadal periods) generally are accompanied by more intense daily precipitation extremes. The introduction indicates that, for the future, the Clausius-Clapeyron equation does suggest that for future warming mean and extreme precipitation should increase. Can we see that relationship already in the past in this simulation ? How large should the 'warm' or cold regions be so that the temperature impacts extreme precipitation (e.g. whole hemispheres or continents) ? If yes, would it be worth to investigate this link in climate reconstructions, as far as proxies for extreme precipitation might be found ?

As we responded in the previous discussion phase, we repeated the GEV analysis using the global mean and regional temperature anomalies as covariates for extreme precipitation and included the result with the new analysis in the revised manuscript in the method (Sect. 3.2, lines 268 - 273) and result (Sect. 4.4, lines 468 - 487) sections. We found that not the global, but the regional surface temperatures are statistically connected with extreme precipitation, mostly over the extratropical land areas and the tropical ocean.

Also note that we modified Fig. 11 and the abstract based on the result of the new analysis.

Lines 268 – 273: "TS is obtained by subtracting the 1501 BCE–1849 CE monthly means of surface air temperature from each monthly value in the simulations. Two kinds of TSs are considered for the non-stationary analysis: one is the globally-averaged means of TS (TS-G; Fig. S4) and another is the spatially (latitude and longitude) gridded TS (TS-R). The former is to assess the influence of global temperature and the latter is to assess the influence of regional temperature on daily extreme precipitation. Both TSs are resolved monthly and similar to the modes of internal variability, they are not interpolated to daily time resolution. The influences of TSs are compared to those of the modes of variability."

Lines 468 – 487: "The result does not vary much when TS-G is included as covariate (Fig. 11b). Only a few sporadic points that denote the influence of TS-G on extreme precipitation appear over the Southern Hemisphere, indicating that the changes in the global mean temperature affect little the long-term variability of extreme precipitation compared to the modes of variability. However, the influence of the surface temperature in Fig. 11b changes when the regional temperature anomalies TS-R instead of TS-G are included as covariates (Fig. 11c). The dominance of TS-R over the modes of variability is highlighted in many land areas and the tropical oceans. The land areas where TS-R is more important are found largely at the extratropical latitudes in the Northern and Southern Hemispheres, covering a large part of northern Asia and North America, southern South America and Africa, Australia, the African transition zone, and the Arabian Peninsula. It is clear that over northern Asia, the influences of PNA and NAM that appeared previously in Fig. 11a and b are masked by those of TS-R. TS-R also prevails over the tropical Pacific, where ENSO takes place. In this region, the values of TS-R overlap with the NIÑO index, which is calculated as an average of the surface temperature anomalies of the same area. Hence, it is reasonable to interpret the predominance of TS-R in the tropical Pacific to be similar to the influence of ENSO. However, this pattern of TS-R simply indicates that the regional temperatures are statistically more associated with the regional extreme precipitation than the averaged temperature condition, thus the NIÑO index, over the area. The here found associations of TS-R with extreme precipitation over land are in line with the preceding studies (Pendergrass et al., 2015; Sillmann et al., 2017; Sun et al., 2021), which have

demonstrated that the present-day and future extreme precipitation are regulated by surface temperature. Here, it is also shown that TS-R does not outperform all the modes of variability over the entire land areas. There are regions, including some in the extratropics, where the modes of variability still play more important roles in regional extreme precipitation. Some of these regions are North America, northern South America, west- and southern Europe, and southern Asia, where ENSO, PDO, EA-WR, and PSA exhibit statistically significant associations."

**Minor comments:**

2) 'Although, eruptions alter both the intensity and frequency of extreme precipitation,.' Change Although to However

We modified the respective sentence in lines 18 – 20 to:

"Tropical volcanic eruptions affect extreme precipitation more clearly in the short term up to a few years, altering both the intensity and frequency of extreme precipitation. However, more apparent changes are found in the frequency than the intensity of extreme precipitation."

*3)* 'Extreme daily precipitation, which often causes devastating flood events, is a complex phenomenon due to its rare occurrence and short-lived nature'

I guess that this are not necessarily the reasons why extreme precipitation is a complex phenomenon. These may, however, be reasons why it is difficult to analyse extreme precipitation.

We modified "complex phenomenon" to "difficult phenomenon to study" in line 26.

4) 'When the sample size is small and the estimated  $\xi$  is negative, there is a bias in the estimation of  $\xi$  towards a larger standard error (Blender et al., 2017)'

This sentence could be a bit confusing at first sight. I guess that here 'sample size' refers to the number of exceedances (?), but some readers would rather interpret 'sample size' as the total number of data (above and below the threshold). In any case, this sentence is not clear to me, since in both cases the 'sample size' would be the same for all grid-cells: either 1% of the available data (threshold set at the 99% percentile) or the all the available data. So why do some grid-cells have a smaller sample size than others?

Thank you for pointing this out. As you said, the "sample size" means the number of exceedances, but after these exceedances being de-clustered to guarantee the statistical independence among the values as mentioned in lines 179 - 182. Although in the beginning, every cell has the same 5% or 1% of their total number of data, the de-clustering method can reduce these numbers at each grid points up to 40%, as shown in Figs. 6 and 7. We changed "sample size" to "number of exceedances" and reformulated the corresponding sentence to clarify errors in negative shape parameters.

Lines 168 - 171: "When the number of exceedances is small, and the estimated  $\xi$  is negative, there is a bias in the estimation of  $\xi$  towards a larger standard error. This occurs because since any sample has a finite maximum, there is a bias towards estimated distributions with an upper limit, hence the negative estimated shape parameter (Giles et al., 2016)."

Lines 183 – 185 : "[...] The result of the de-clustering can be quantified through an extremal index, which is the ratio between the number of extremes after being de-clustered and the initial number of extremes. These de-clustered exceedances are the values used for the analysis."

---

## Author Response (AR2)

Manuscript cp-2021-61

Dr. Ran Feng, editor Climate of the Past

Woon Mi Kim
Physics Institute and Oeschger Centre for Climate Change Research, University of Bern
Sidlerstrasse 5, 3012 Bern, Switzerland
September 17, 2021

Dear Dr. Ran Feng,

Thanks very much for considering our manuscript "Statistical characteristics of extreme daily precipitation during 1501 BCE – 1849 CE in the Community Earth System Model" for the publication in the Climate of the past.

As you requested, we made the CESM 1.2.2 data used for the analysis publicly available in Zotero with an assigned DOI 10.5281/zenodo.5513689 (https://doi.org/10.5281/zenodo.5513689). The analysis is based on the two R packages, ismev (https://CRAN.R-project.org/package=ismev) and extRemes (https://CRAN.R-project.org/package=extRemes; Gilleland and Katz, 2016) that are already publicly available. We uploaded a code to generate Fig. 11 (Fig. 10 follows the similar calculation procedure to Fig. 11.), which involves the calculation of D statistics and the comparison among the estimated D values, on the personal GitHub https://github.com/wmk21/EVT-D-statistics-comparison. We mentioned these directories in the data and code availability sections of the manuscript.

Thanks again for your dedication to our manuscript,

Sincerely,

Woon Mi Kim, on behalf of all the authors